# Hyper-Production of Pullulan by a Novel Fungus of *Aureobasidium melanogenum* ZH27 through Batch Fermentation

**DOI:** 10.3390/ijms25010319

**Published:** 2023-12-25

**Authors:** Qin-Qing Wang, Jia Lin, Qian-Zhi Zhou, Juan Peng, Qi Zhang, Jiang-Hai Wang

**Affiliations:** 1Guangdong Provincial Key Laboratory of Marine Resources and Coastal Engineering, School of Marine Sciences, Sun Yat-Sen University, Zhuhai 519082, China; linjia26@mail.sysu.edu.cn (J.L.); zhouqzh6@mail.sysu.edu.cn (Q.-Z.Z.); pengj28@mail.sysu.edu.cn (J.P.); zhangq528@mail2.sysu.edu.cn (Q.Z.); 2Guangdong Engineering Laboratory of Biomass High-Value Utilization, Institute of Biological and Medical Engineering, Guangdong Academy of Sciences, Guangzhou 510316, China

**Keywords:** pullulan, *Aureobasidium melanogenum*, gene expression, swollen cells, vacuoles

## Abstract

Pullulan, which is a microbial exopolysaccharide, has found widespread applications in foods, biomedicines, and cosmetics. Despite its versatility, most wild-type strains tend to yield low levels of pullulan production, and their mutants present genetic instability, achieving a limited increase in pullulan production. Therefore, mining new wild strains with robust pullulan-producing abilities remains an urgent concern. In this study, we found a novel strain, namely, *Aureobasidium melanogenum* ZH27, that had a remarkable pullulan-producing capacity and optimized its cultivation conditions using the one-factor-at-a-time method. To elucidate the reasons that drove the hyper-production of pullulan, we scrutinized changes in cell morphology and gene expressions. The results reveal that strain ZH27 achieved 115.4 ± 1.82 g/L pullulan with a productivity of 0.87 g/L/h during batch fermentation within 132 h under the optimized condition (OC). This pullulan titer increased by 105% compared with the initial condition (IC). Intriguingly, under the OC, swollen cells featuring 1–2 large vacuoles predominated during a rapid pullulan accumulation, while these swollen cells with one large vacuole and several smaller ones were prevalent under the IC. Moreover, the expressions of genes associated with pullulan accumulation and by-product synthesis were almost all upregulated. These findings suggest that swollen cells and large vacuoles may play pivotal roles in the high level of pullulan production, and the accumulation of by-products also potentially contributes to pullulan synthesis. This study provides a novel and promising candidate for industrial pullulan production.

## 1. Introduction

Pullulan is a valuable and renewable exopolysaccharide (EPS) synthesized by the yeast-like fungi *Aureobasidium* [1]. It comprises repeating maltotriose subunits (three glucoses linked via two α-(1→4) glycosidic bonds) linked via α-(1→6) glycosidic bonds [2], which impart unique physico-chemical properties, including water solubility, adhesive ability, structural flexibility and elasticity, film-forming ability, oxygen impermeability, biocompatibility, and biodegradability [3,4]. Most yeasts with the ability to produce pullulan are generally regarded as safe (GRAS) [4,5]. Notably, pullulan has gained approval as an ingredient in foods and drugs in Japan, USA, Europe, and China, and as a fresh-keeping agent and stabilizer [4]. Moreover, pullulan-based capsule shells are also considered safe and manufactured by reputable companies, such as Hayashibara Co., Ltd. (Okayama, Japan) and Shandong Kangnaxin Biotechnology Co. Ltd. (Weifang, China) [4]. Nowadays, pullulan and its derivatives have extensive applications in various sectors, including food, packaging, biomedicine, cosmetics, and environmental remediation [6]. Consequently, pullulan has emerged as a multifunctional EPS with numerous benefits for human well-being and the environment.

*Aureobasidium* spp. strains are well known to be the primary producers of pullulan. The strains are multicellular fungi encompassing yeast-like cells, swollen cells, conidia, chlamydospores, and mycelia types that thrive in diverse habitats, such as soils, plant leaves, oceans, and extreme environments [7,8,9]. Among the species of *Aureobasidium*, *A. pullulans* and *A. melanogenum* emerge as the most robust pullulan producers [1,10,11]. However, most wild-type strains yield less than 78 g/L of pullulan [2,3,8,12]. Only a few wild isolates were found to produce relatively high pullulan titers (>100 g/L). For instance, *A. melanogenum* A4 using maltose achieved 122.3 g/L pullulan with a yield of 0.4 g/g via fed-batch fermentation and another strain *A. melanogenum* TN1-2 produced 114 g/L pullulan from sucrose with 0.81 g/g yield via batch fermentation [10,13]. Some engineered mutants also exhibited distinctly elevated pullulan production, such as the mutant *A. pullulans* M233-20, which produced 162.3 g/L pullulan [14]. To improve pullulan production, various measures, including condition optimization, mutagenesis, genetic modification, and substance addition, were employed [9,15,16]. However, the improvements in pullulan production achieved using these measures were limited to no more than 35%, which, in return, impacted the molecular weight of pullulan [1,15,17]. Hence, the above-mentioned measures are still insufficient to meet the requirements of efficient and high-quality industrial pullulan production. On the other hand, high pullulan production requires high sugar concentrations, which increase the osmotic stress and actually make osmophilic or osmotolerant strains more suitable for pullulan production [18].

Most *Aureobasidium* strains prefer to use sucrose as the best carbon source for pullulan production, while only a small amount of them prefer utilizing glucose, fructose, or maltose [2,10,18,19]. For example, *A. melanogenium* P16 produced the highest pullulan titer (65.3 g/L) using sucrose in comparison with glucose, fructose, glycerol, and inulin [12]. However, in addition to pullulan production, *Aureobasidium* strains have the ability to convert sucrose into several by-products, such as fructooligosaccharides (FOSs), trehalose, and glycerol, which contribute to mediate osmotic stress and cell survival [13,20]. Yet, the accumulation of these by-products reduces the carbon metabolic flux toward pullulan synthesis, potentially reducing pullulan yield [13,20].

To meet the demands for industrial production and applications of pullulan, it is imperative to possess wild-type strains that exhibit robust pullulan-producing capabilities. However, the availability of such strains remains limited. This study focused on systematically investigating a newly discovered strain ZH27, which showed a high-level EPS-producing capacity and was suspected to be *Aureobasidium melanogenum*. The conditions for EPS production were optimized using the one-factor-at-a-time method. Subsequently, the structure of purified EPS was identified through enzyme hydrolysis, thin-layer chromatography (TLC), and Fourier-transform infrared spectroscopy (FTIR). Furthermore, we sought to understand the reasons behind the high levels of pullulan production by strain ZH27, which were elucidated by examining changes in cell morphology and gene expressions associated with the syntheses of pullulan and by-products (FOSs, trehalose, and glycerol).

## 2. Results

### 2.1. Screening and Identification of EPS-Producing Strains

Microorganisms capable of producing EPS are ubiquitously distributed in diverse environments due to the protective properties of EPS and various secreted enzymes for degrading organics for growth. In this study, a novel strain ZH27 sharing features reminiscent of *Aureobasidium* strains was screened through colony characteristics. Strain ZH27 produced a remarkable EPS production of 40 ± 1.2 g/L during the initial screening phase, surpassing the levels observed in two other isolates (30–33 g/L). *Aureobasidium* strains exhibit varied morphological features under different conditions [9]. When cultivated on YPD (1.0% yeast extract, 2.0% peptone, 2.0% glucose, and 1.5% agar) plates for 2.5 days, the colonies of strain ZH27 developed gradually into light pink, sticky, smooth, and round structures with radial mycelia, which were surrounded or coated by EPS (Figure 1A(a)). By the fifth day, the colonies underwent a transition into non-tacky and thick textures with a milk-white feature (Figure 1A(b)). In contrast, the colonies on the potato dextrose medium (PDA) plates displayed a predominantly white appearance with radial mycelia for 2.5 days (Figure 1A(c)). Subsequently, the color of the colonies turned into a yellowish-brown, culminating in complete black (Figure 1A(d,e). Furthermore, three distinct cell types (yeast-like cells, blastospores, and mycelia) of strain ZH27 were observed in the liquid YPD (Figure 1A(f)).

To identify strain ZH27, the phylogenetic tree was constructed using neighbor-joining analysis based on the internal transcribed spacer (ITS) sequences. As depicted in Figure 1B, strain ZH27 (GenBank no. MN658859) shared a phylogenetic similarity of 99.82% with *A. melanogenum* strains ATCC 9348 and CBS 123.37. Both morphological characteristics and molecular identification strongly support the classification of strain ZH27 as one member of the *A. melanogenum* species.

### 2.2. Optimization for EPS Production

The medium composition is a crucial factor that influences the yield of EPS [7]. Therefore, the optimization was performed for strain ZH27 to improve the EPS titer in this study. Figure 2a reveals that strain ZH27 using sucrose achieved the highest EPS titer of 40 g/L and simultaneously the lowest dried cell weight (DCW) among all the carbon sources (sucrose, glucose, fructose, maltose, and glycerol). Moreover, the EPS titers increased with sucrose concentrations rising up to 15%, after which they declined (Figure 2b), while the corresponding DCWs of strain ZH27 showed insignificant changes within the range of sucrose concentrations (13–17%) (Figure 2b). Clearly, the results indicate that sucrose emerged as the optimal carbon source for EPS production by strain ZH27, yielding the highest EPS titer of 49 ± 1.17 g/L at 15% sucrose.

Ammonium sulfate ((NH_4_)_2_SO_4_) depletion is known to induce EPS synthesis in microorganisms [7]. The EPS titers decreased continually with increasing (NH_4_)_2_SO_4_ concentrations, while the DCWs exhibited an increase followed by a decrease beyond 0.12% (NH_4_)_2_SO_4_ (Figure 2c). This suggests nitrogen catabolite repression (NCR) appeared in EPS synthesis by strain ZH27 after 0.12% (NH_4_)_2_SO_4_, and 0.01% (NH_4_)_2_SO_4_ was identified as the preferred concentration for EPS production. Figure 2d shows that higher EPS titers were achieved at 0.3–0.7% K_2_HPO_4_ compared with 0.1% K_2_HPO_4_ and remained relatively stable, mirroring biomass changes. This suggests that 0.3% K_2_HPO_4_ is optimal for EPS synthesis by strain ZH27, indicating a relative insensitivity to changes in K_2_HPO_4_ concentrations. Strain ZH27 from a mangrove ecosystem exhibited a degree of salt stress tolerance [21]. Improved growth of strain ZH27 was observed at 0.1–0.5% NaCl compared with no NaCl, with DCWs declining below 0.4% NaCl but peaking at 0.5% NaCl (Figure 2e). Simultaneously, the highest EPS titer of 67.8 ± 2.5 g/L was achieved at 0.4% NaCl (Figure 2e).

The cell growth and cellular metabolic capacity of strain ZH27 were influenced by the culture conditions. Figure 2f shows that EPS titers and biomass of strain ZH27 all exhibited upward trends as the temperature increased to 28 °C and then declined, indicating that 28 °C was the most favorable temperature for strain ZH27 to produce EPS (68.5 ± 0.63 g/L). Additionally, strain ZH27 achieved the maximum EPS titer (108 ± 2.2 g/L) at an initial pH of 5, associating with insignificant changes in biomass beyond the optimal pH value (Figure 2g).

Consequently, the final enhanced medium (EM) comprised 15% sucrose, 0.3% yeast extract, 0.3 % K_2_HPO_4_, 0.02% MgSO_4_·7H_2_O, 0.4% NaCl, and 0.01% (NH_4_)_2_SO_4_, with an initial pH of 5 at 28 °C.

### 2.3. Structural Analysis of EPS

The analysis of the EPS structure involved pullulanase hydrolysis, TLC, and FTIR. Pullulanase specifically hydrolyzes pullulan to produce maltotriose [14]. After purification, the purified EPS, when hydrolyzed by pullulanase, released a significantly higher quantity of reducing sugars, increasing by 110.7-fold compared with the control. In contrast, the reducing sugars from a commercial pullulan increased by 3.83-fold (Figure 3a). Moreover, the migration heights of components in hydrolysates from the purified EPS and commercial pullulan on a TLC plate matched that of maltotriose (1%) (Figure 3b), affirming the presence of maltotriose in the purified EPS. These results indicate that the EPS produced by strain ZH27 was pullulan.

Further confirmation of the structure and elemental composition of the purified EPS was obtained via FTIR analysis. The FTIR spectrum of the purified EPS closely resembled that of a commercial pullulan (Figure 3c). The absorption peak at 3423.03 cm^−1^ for the purified EPS was close to that (3396.03 cm^−1^) of commercial pullulan (Appendix A), signifying the presence of the stretching vibration of hydroxyl groups (–OH) in the EPS [2]. The signal near 2923.60 cm^−1^ from both samples was attributed to the stretching vibration of the C–H bonds in methylidyne and methylene of sugar rings [10]. Additionally, the characteristic bonds O–C–O, C–O–C, C–O, and C–O–H in the purified EPS were identified on the basis of absorption peaks near 1637.29 cm^−1^, 1159.03 cm^−1^, 1025.96 cm^−1^, 1427.09 cm^−1^, and 1371.16 cm^−1^, respectively (Appendix A). Notably, the presence of α-1,6-D-glucosidic linkages, α-D-glucopyranose, and α-1,4-glucosidic linkages was verified by the absorption peaks at 933.38 cm^−1^, 852.40 cm^−1^, and 755.97 cm^−1^, respectively. The functional groups in the EPS closely resembled those in the commercial pullulan, providing additional confirmation that the EPS derived from strain ZH27 was indeed pullulan.

### 2.4. Comparison of Pullulan Production between the IC and OC

To evaluate the pullulan produced by strain ZH27, cultivations were scaled up to 7 L under the initial condition (IC) and optimized condition (OC). In order to enhance the pullulan titer, the use of YPS medium (using sucrose to replace glucose in YPD) was established as the optimal seed medium for incubating strain ZH27 for 36 h with a suitable inoculum size of 5% for fermentation (Appendix A). Figure 4 illustrates that pullulan titers under the OC experienced a rapid accumulation after 36 h, reaching a maximum of 115.4 ± 1.82 g/L with a yield of 0.77 g/g and a productivity of 0.87 g/L/h at 132 h. This pullulan titer was more than twice compared with that under the IC. Total sugars under the OC were utilized more rapidly and more thoroughly than those under the IC, leaving the total sugar content of 9.6% at 132 h (Figure 4). The consumption rate of total sugars under the OC within 132 h was 1.03 g/L/h, which was higher than the rate of 0.67 g/L/h under the IC (Figure 4). Interestingly, the reducing sugars maintained relatively high levels under the OC within the first 60 h but were subsequently consumed more rapidly and more completely compared with those under the IC (Figure 4). There were slight increases in DCWs and growth rates under the OC than those under the IC after 60 h (Figure 4). The DCW (21.4 ± 0.3 g/L) and growth rate (0.16 g/L/h) of strain ZH27 under the OC at 132 h were slightly higher than those under the IC (DCW, 20.4 ± 0.2 g/L; growth rate, 0.15 g/L/h). These results indicate that the cells of strain ZH27 exhibited robust growth and high pullulan production under high osmotic stress. Notably, the final supernatant viscosity at 132 h significantly decreased to 306 mPa·s under the OC compared with that (10,753 mPa·s) under the IC (Appendix A). This highlights that the optimization of conditions for pullulan production by strain ZH27 can remarkably enhance its production level.

### 2.5. Probable Reasons for Hyper-Production of Pullulan

Compared with the pullulan titer under the IC, higher pullulan production was achieved by strain ZH27 under the OC. We considered that the OC with a higher concentration of sucrose and a lower concentration of (NH_4_)_2_SO_4_ was characterized by a high osmotic stress and a high ratio of carbon to nitrogen (C/N). These conditions are likely to trigger self-regulation in the cells of strain ZH27, influencing multiple aspects, such as cell morphology and gene expressions, to adapt to the environment under the OC.

#### 2.5.1. Dynamic Changes in Cell Morphology during Batch Fermentation

*Aureobasidium* fungus displayed various cell types during the fermentation process. As depicted in Figure 5, the cells of strain ZH27 under both the OC and IC primarily developed into yeast-like cells, arthroconidia, and mycelia at the initial stage of fermentation (12–48 h). However, a significant difference emerged between the two conditions beyond this point. Under the OC (48–132 h), cells further differentiated into dominant swollen cells with a small amount of yeast-like cells. In contrast, swollen cells were also predominant under the IC during the same period but only accompanied by a small number of yeast-like cells, arthroconidia, and mycelia. Ultimately, chlamydospores formed earlier (96 h) under the IC compared with the OC (144 h). Notably, most of the swollen cells under the OC contained 1–2 large vacuoles but the swollen cells with one vacuole were dominant after 48 h, whereas swollen cells under the IC had one large vacuole and several small ones (Figure 5).

Previous studies suggested that yeast-like cells can differentiate into swollen cells via the regulation of low pH values (<6.0) [22]. Figure 5 shows that the hyper-osmotic stress and low pH values (ultimately 3.57) (Appendix A) under the OC prompted the cells of strain ZH27 to form swollen cells earlier and more extensively than those under the IC. This resulted in an evident difference in vacuoles within swollen cells and delayed the formation of chlamydospores. However, a previous study proposed that the arthroconidia of *A. melanogenum* TN3-1 and numerous small vacuoles might allow for withstanding high osmotic stress and producing high pullulan titers [18]. This finding contrasts with our observations under the OC.

#### 2.5.2. Expression Changes of the Genes in Pullulan Synthesis and Carbon Metabolism

Compared with the transcriptional gene expression of strain ZH27 under the IC, quantitative real-time PCR (qPCR) tests revealed significant transcriptional changes in its genes involved in carbon metabolism and pullulan synthesis under the OC. During the stage where sucrose entered the cells, three genes encoding glucose transporters and *SUC2* encoding the sucrose-proton symporter were significantly upregulated under the OC (Table 1). Most genes associated with carbon metabolism, pullulan synthesis and regulation showed upregulated expressions, including *GluK* encoding glucose kinase, *UGP* encoding UDPG-pyrophosphorylase, *UGT1* encoding UDP-glucosyltranferase, *Ags2* encoding α-glucan synthase 2, and *PUL1* encoding pullulan synthetase, which were upregulated by 18.4-, 2.15-, 46.4-, 3.7-, and 4.4-fold, respectively (Table 1). Additionally, the transcriptional levels of glycolipid transfer protein genes *Apo* and *Gltp* were also upregulated by 1.54- and 3.78-fold, respectively (Table 1). These findings indicate enhanced gene expression levels in most steps of the proposed pullulan synthesis pathway.

Moreover, the expressions of genes related to the synthesis of by-products and metabolism also underwent some changes. FOS synthesis was likely intensified due to upregulated expressions of fructoslytranserase genes (FFases) encoding fructoslytranserases by 2.2–3.7-fold, and their transcription factor gene *FTR1* was upregulated by 3.3-fold (Table 1). Similarly, the transcriptional levels of the α-trehalose-6-phosphate synthase gene (*TPS1*) and trehalose-6-phosphate phosphatase gene (*TPP*) responsible for trehalose synthesis were upregulated by 5.0-, and 2.8-fold respectively, potentially strengthening the trehalose synthesis and UDPG consumption (Table 1). Additionally, two key genes of glycerol-3-phosphate dehydrogenase gene (*GPD*) and glycerol-3-phosphatase gene (*GPP*) required for glycerol synthesis were upregulated by 3.8- and 13.4-fold, respectively, and probably improved the glycerol synthesis (Table 1). Interestingly, high expressions (2.0–18.4-fold) of genes encoding three rate-limiting enzymes (glucose kinase (GluK), phosphofructo-kinase (PFK), and pyruvate kinase (PYK)), glyceraldehyde-3-phosphate dehydrogenase (G3P), and phosphoglycerate kinase (PGK) efficiently accelerated glycolysis and generated more pyruvate (Table 1). Subsequently, acetyl-CoA could be converted from pyruvate by the pyruvate decarboxylase gene (*PYD*), which was upregulated by 2.45-fold.

Various regulatory factors play crucial roles in responding to high osmotic stress and regulating metabolite synthesis [1]. The glucose repressor CreA, which is a key protein in carbon catabolite repression, was upregulated by 2.23-fold in strain ZH27 under the OC and potentially suppressed pullulan synthesis (Table 1). Moreover, two GATA factors (AreA activator and AreB repressor) with important effects on NCR exhibited changes under the OC. The upregulation of *AreA* (1.36-fold) and downregulation of *AreB* (0.85-fold) under the OC seemed to facilitate pullulan synthesis (Table 1). In the pathway of the high osmolarity glycerol mitogen-activated protein kinase (HOG-MAPK), the expressions of *Hog1* (2.60-fold) and high osmolarity signaling factor *Sho1* (1.60-fold) were all upregulated (Table 1), contributing to balancing osmotic stress by regulating the expressions of osmoresponsive genes [13,23]. In addition, Msn2, which is a global regulator, was also distinctly upregulated in strain ZH27 under the OC by 2.29-fold.

## 3. Discussion

Pullulan, which is a versatile and valuable EPS produced by *Aureobasidium* strains, has diverse applications that benefit both health and the environment [4]. Some wild-type *Aureobasidium* strains can achieve relatively high pullulan levels (>100 g/L) [1], while the majority of them are limited to low levels of pullulan production (≤78 g/L) [3,8,12]. Various approaches, including strain screening, mutagenesis, and genetic modification, have been explored to enhance pullulan production [1,9,14]. However, these methods come with challenges, such as genetic instability and strain degeneration due to genomic rearrangements [24]. The genetic instability of mutants [24] and their limited increase in pullulan titers [1,25] have posed challenges for industrial pullulan production, emphasizing the urgent demand for wild-type strains with high pullulan-producing abilities. In this study, we identified and investigated promising target strains with the potential for high pullulan production, confirming the EPS produced as pullulan. Subsequently, we delved into changes in cell morphology and gene expressions during pullulan synthesis after optimizing conditions.

### 3.1. Strain ZH27 Is a Promising Candidate for Pullulan Synthesis

The target strain ZH27, while producing 40 g/L EPS, was obtained during the initial screening step. In comparison with EPS titers (<30 g/L) derived from the initial screening step in previous studies [2,10], strain ZH27 is considered to produce a relatively high EPS titer. Moreover, it is even comparable to the EPS titer (45.7 g/L) produced by *A. melanogenum* 13-2 at 96 h [8], suggesting that strain ZH27 may be a potential candidate for EPS production. It exhibited multiple cell types and variable colony colors (Figure 1A), which are quite similar to the features of *Aureobasidium* strains [8,9]. As well-known black yeasts, *Aureobasidium* strains consist of five cell morphologies and synthesize several products (e.g., pullulan, polymalate, and biosurfactant) in response to environmental stress [26,27]. These observations indicate that strain ZH27 may belong to one member of *A. melanogenum* (Figure 1B). However, the colonies of strain ZH27 surrounded by radial mycelia displayed different characteristics compared with the colonies of other *A. melanogenum* strains, which were covered by EPS but lacked radial mycelia [8,13].

Pullulan is composed of maltotriose units connected by α-(1→6) glycosidic bonds, which can be specifically hydrolyzed by pullulanase. Various methods, including TLC, FTIR, and nuclear magnetic resonance spectroscopy, have been employed to determine the molecular structure of pullulan [2,19]. In this study, after optimization (discussed in the following part), the EPS produced by strain ZH27 was hydrolyzed by pullulanase, generating maltotriose and glucose (Figure 3a,b), which is consistent with previous results [10,13]. Furthermore, FTIR analysis revealed that the EPS from strain ZH27 exhibited functional groups, such as O–C–O (1637.29 cm^−1^) and C–O–C (1159.03 cm^−1^), and characteristic α-1,6-glucosidic and α-1,4-glucosidic linkages, which closely resembled those present in a standard pullulan (Appendix A). These results are also in agreement with previous findings [2,3,28].

### 3.2. Robust Pullulan Synthesis by Strain ZH27 after Optimization

Pullulan titers generally depend on high concentrations of carbon sources [29]. In the quest for high pullulan production, we optimized several key factors. Notably, strain ZH27 exhibited a preference for 15% sucrose as the optimal carbon source, which is a concentration rarely reached by wild-type strains (Figure 2a,b). Interestingly, except for mutant M233-20, which evolved from whole-cell mutagenesis to be able to convert 20% glucose into pullulan [14], 15% sucrose was the highest concentration of single sugar (Figure 2b) used to produce pullulan by wild-type strains so far [1,7]. It was found that the wild-type strain ZH27 showed tolerance to hyper-osmotic stress. Its pullulan accumulation was significantly and negatively influenced by (NH_4_)_2_SO_4_ concentrations, which is sensitive to (NH_4_)_2_SO_4_ concentrations due to occurrence of NCR at 0.12% (NH_4_)_2_SO_4_ (Figure 2c), while *A. pullulans* CCTCC M201225 and *A. melanogenum* P16 exhibit NCR at 0.24–0.30% (NH_4_)_2_SO_4_ [30,31]. Thus, a low concentration of (NH_4_)_2_SO_4_ (0.01%) was found to be preferable for pullulan production, redirecting carbon flux toward metabolite synthesis rather than biomass formation by activating some enzymes and enhancing the ATP supply [31].

K_2_HPO_4_, which is a crucial factor known to influence pullulan production [19,32], did not significantly impact strain ZH27 to produce pullulan, highlighting differences in sensitivities between different strains and their regulation to the genes involved in pullulan synthesis (Figure 2d) [2,13]. The pullulan production of strain ZH27 was also sensitive to changes in NaCl concentrations, with the highest pullulan titer (67.8 ± 2.5 g/L) observed at 0.4% NaCl and an increase of 11% compared with the control at 0.1% NaCl (Figure 2e). This change is similar to a previously reported increase of 26.7% at 0.3% NaCl [33]. Interestingly, the highest biomass of strain ZH27 occurred at 0.5% NaCl, likely due to the formation of small molecules (e.g., glycerol) for self-protection against high salinity and osmotic stress. Consequently, enhancing carbon flow toward biomass replaced pullulan synthesis [34]. Moreover, high expressions of some enzymes (e.g., amylases) induced by high NaCl concentrations probably contributed to the reduction in pullulan titers [33]. These findings highlight the complicated interplay of factors influencing pullulan production in strain ZH27.

Many *Aureobasidum* strains optimally produce pullulan within the temperature range of 24–30 °C [6]. Nevertheless, a temperature suitable for cell growth may not always favor metabolite synthesis. For instance, *A. pullulans* CGMCC1234 grew well at 32 °C but favored producing pullulan at 26 °C, implying that a two-step temperature regulation might enhance pullulan production [35]. Interestingly, Hilares et al. (2019) reported that the optimal temperature for both cell growth and pullulan synthesis of *A. pullulans* LB83 was the same (25 °C) [36]. Hence, temperature is considered an important regulator for pullulan synthesis. Indeed, the pullulan yield was remarkably restricted at lower initial pH values (<5.0) or higher ones (>7.0), indicating that initial pH values sensitively influence the pullulan-producing process of strain ZH27 (Figure 2g). At different pH values, pullulan-producing strains may exhibit specific cell types that are likely required for pullulan synthesis. These findings emphasize the importance of precise environmental conditions for efficient pullulan synthesis.

Recently, a few wild-type strains have achieved pullulan levels over 100 g/L (Table 2). Several *A. melanogenum* strains can produce high pullulan titers (110–122.3 g/L), but some of them have very low yields (<0.5 g/g) [10,13,18]. Therefore, it is still urgent to mine novel strains to produce pullulan with high titer, high yield, and high productivity. After optimization, strain ZH27 exhibited a remarkable pullulan production capability, achieving a pullulan titer of 115.4 ± 1.82 g/L at 132 h and an improvement of 105% under the OC compared with the IC (Figure 4). Simultaneously, strain ZH27 exhibited a high pullulan yield (0.77 g/g), productivity (0.87 g/L/h), and DCW (21.4 g/L) through batch fermentation (Figure 4). Notably, more than 90% of sugars were utilized for pullulan synthesis and cell growth, reflecting a high efficiency compared with the reported pullulan titers (48.9–66.8 g/L) produced by *A. pullulans* strains RBF-4A3 and MR using 15% glucose or sucrose within 144 h [29,37]. This implies that the efficient pullulan synthesis of strain ZH27 may be attributed to its good adaptability to high osmotic tolerance. The pullulan yield (0.77 g/g) of strain ZH27 using sucrose was close to the reported one (~0.80 g/g) from high pullulan-producing wild-type *A. melanogenum* strains [13,18]. As for the productivity (0.87 g/L/h), it achieved a high level via batch fermentation among the wild-type strains, even though it was lower than those from fed-batch fermentation or some mutants [10,11,13,14]. Notably, the lower supernatant viscosity under the OC might result from high expressions of some enzymes (e.g., α-amylase and pullulanase) that hydrolyze pullulan (Table 2), just as reported by a previous study [38]. More importantly, pullulan produced by strain ZH27 was free of melanin during fermentation (Appendix A), making it a promising candidate for the industrial production of melanin-free pullulan.

### 3.3. Swollen Cells, By-Products, and High Expressions of Related Genes May Be Required for Efficient Pullulan Production

Several cell types may coexist in a certain cultivation period, while the rest may not. The differentiation of cells in strain ZH27 under the OC revealed a shift toward mainly swollen cells after 48 h, contrasting with the control under the IC. The occurrence of chlamydospores was minimal within 132 h under the OC (Figure 5), and most swollen cells during this period contained 1–2 large vacuoles, which differed from the main single large vacuole with several small ones in the control (Figure 5). These changes in cell differentiation and vacuole characteristics were likely influenced by variations in nutrients and pH values, impacting pullulan synthesis. Swollen cells emerged as a prominent cell type during pullulan synthesis in *A. pullulans* NG [40], exhibiting a robust stress defense and antioxidative capabilities compared with yeast-like cells [22]. As shown in Figure 5, swollen cells under the OC seemed to have a stronger and more stable ability to withstand environmental stress, which may require for high levels of pullulan production. Simultaneously, the number and volume of vacuoles in the swollen cells of strain ZH27 are speculated to play a crucial role in pullulan synthesis. The distinct features of vacuoles in the swollen cells of strain ZH27 compared with strain TN3-1 may account for the resistance to high osmotic stress and high levels of pullulan production [18]. 

It was found that *A. pullulans* NG grew well under the nutrient-rich condition but produced mycelia and chlamydospores under the nutrient-poor condition; while the osmotic stress prevented the differentiation of yeast-like cells into two types of the above-mentioned cells [41]. Both one type of cells (e.g., yeast-like cells [13], swollen cells [40], or arthroconidia [18]) and two types of cells (e.g., yeast-like cells and swollen cells) [9] were conducive to pullulan synthesis. However, the exact cell type responsible for pullulan synthesis in *Aureobasidium* strains remains unclear [1]. Moreover, the pH is identified as another factor that influences cell morphology and pullulan yield [42]. Yeast-like cells as core cells mainly occurred at the beginning of cultivation with higher pH values (≥6.0), while low pH values induced the differentiation of yeast-like cells into swollen cells [22]. The regulation of pH values seems to be a crucial factor in determining the cell types during the cultivation period.

Under high osmotic stress, genes related to the syntheses of pullulan and by-products in strain ZH27 underwent some significant changes [1] (Figure 6). First, enhanced expressions of glucose transporters and SUC2 under the OC suggest an efficient transport of sucrose and newly generated glucose into cells, which agrees with the way sucrose enters the cells of *A. pullulans* for polymalate synthesis [43]. Then, all of the genes involved in pullulan synthesis were upregulated except PGM under the OC, which contributed to the high level of pullulan production (Table 1). For instance, the high expressions of *Apo* and *Gltp* (1.54–3.78 folds) responsible for the formation of Lph-glucose seemed to promote pullulan synthesis (Table 1). These observations are consistent with the previous report stating that the over-expression of *Apo* resulted in an increase of ~10% in pullulan titer [44]. The upregulated expression of *Ags2* (3.7-fold) would probably accelerate the formation and transportation of pullulan precursors and facilitate pullulan synthesis [15]. During pullulan synthesis, UDPG transformed by *UGP* could also synthesize several products (e.g., trehalose and glycogen), but it was the sole precursor for pullulan synthesis. Consequently, the upregulation of the *UGP* expression (2.2-fold) probably facilitated the efficient pullulan synthesis by strain ZH27 [2].

During pullulan synthesis under high sucrose concentrations, *Aureobasidium* strains also produce by-products, mainly including FOSs, trehalose, and glycerol [20,45]. FOSs synthesized by *Aureobasidium* cells using sucrose were considered to lower its extracellular osmotic stress and pullulan titer [46]. However, FOSs would be subsequently hydrolyzed at late stages, contributing to pullulan accumulation [47]. The ability of strain ZH27 to hydrolyze FOSs could be attributed to the high expressions of hydrolases (e.g., α-amylase, glucoamylase, and pullulanase) (Table 1). The hydrolyzed FOSs increased pullulan titers but reduced pullulan viscosity or molecular weight (Appendix A), suggesting an efficient utilization of FOSs. These changes are consistent with the findings in previous report [47]. On the other hand, trehalose and glycerol are crucial osmolytes for intracellular osmotic stress alleviation [48]. They might be highly accumulated in the cells of strain ZH27 under the OC because of significantly upregulated expressions of its related genes (Table 1). This is similar to the upregulations of three genes (*GPP*, *TPS1*, and *TPP*) in strain TN3-1 (110 g/L pullulan) in comparison with strain P16 (65.3 g/L pullulan) [18]. These findings demonstrate that the strains of *Aureobasidium* producing high levels of pullulan contained more glycerol and trehalose compared with those with a low pullulan-producing ability. However, their high expressions could divert the carbon flux away from pullulan synthesis. Consequently, a delicate balance may be maintained to ensure the optimal pullulan yield and simultaneously facilitate the production of necessary by-products. Furthermore, both pullulan and melanin secreted by *Aureobasidium* cells are known to resist high osmotic stress [9]. More interestingly, melanin was not observed during the whole process of pullulan synthesis by strain ZH27 under the OC.

The synthesis of metabolites (such as pullulan) and cell growth of *Aureobasidium* strains are energy-intensive processes [16]. The upregulation of key enzymes (e.g., GluK, PFK, PYK, and PYD) involved in glycolysis in strain ZH27 under the OC (Table 1) indicates an enhanced glycolytic activity, potentially providing more acetyl-CoA for the tricarboxylic acid (TCA) cycle. This metabolic shift supports the energy-intensive processes of pullulan synthesis, cell growth, and by-product accumulation under the OC. Nevertheless, it contrasts with the effect of zinc sulfate addition, where the metabolic flux shifted away from glycolysis and the TCA cycle [49].

Pullulan synthesis, which is sensitive to high carbon source concentrations (Figure 2b), is suppressed by the glucose repressor CreA. Interestingly, the *CreA* gene was upregulated under the OC (Table 1), whereas pullulan production was exactly improved (Figure 4). By contrast, when *CreA* was downregulated in *A. melanogenum* strains, the pullulan production was over 100 g/L [13,18]. This difference may probably be due to evolutionarily genetic variations and different tolerances of strains to high osmotic stress. Therefore, we believe that strain ZH27 may produce a significantly higher pullulan titer under the OC after deleting *CreA*. AreA and AreB play direct roles in NCR and regulating pullulan synthesis [31]. The upregulation of *AreA* and downregulation of *AreB* in strain ZH27 under the OC indicate an efficient nitrogen utilization due to high C/N ratios under the OC compared with the IC (Figure 2b,c), which is consistent with those observed in strain P16 [31]. The HOG-MAPK pathway has been considered a central signaling pathway for responding to high osmotic stress in yeast cells [50]. Hog1, which is a crucial component in the HOG-MAPK pathway, could mediate the expressions of numerous osmo-responsive genes (e.g., *GPD* and *GPP*) and simultaneously adjust their enzyme activities [13,23]. Sho1, which is located in the plasma membrane, is also required for activating the HOG-MAPK pathway and responding to osmotic stress [34]. Therefore, the upregulated expressions of *Hog1* and *Sho1* likely favored to resist osmotic stress and enhanced glycerol synthesis by strain ZH27 (Table 1). Additionally, the regulator Msn2 can upregulate more than 600 genes, including the genes related to pullulan synthesis (e.g., *UGT1*, *UGP1*, and *Ags2*), glycerol accumulation, glycolysis, and the TCA cycle under a stressful environment [51]. The upregulation of gene *Msn2* in strain ZH27 under the OC may account for the high level of pullulan production to a certain extent.

## 4. Materials and Methods

### 4.1. Sample Collection and Screening for EPS-Producing Strains

Leave samples were collected from the mangrove ecosystem of Qi’ao Island in Guangdong, China (N22°23′, E113°38′). The screening was conducted according to the previously reported method with a slight modification [2]. Briefly, the samples were first soaked in a shake flask (250 mL) with 30 mL distilled water for 2 days at 25 °C. The prepared solution (100 μL) was then transferred into a sterile centrifuge tube (50 mL) with minimal salt medium (MSM, 10 mL) and 50 mg/mL chloramphenicol. After being cultivated at 180 rpm and 25 °C for 2 days, the culture (20 μL) was spread on MSM plates and cultivated at 25 °C. Each colony was purified three times through YPD plates. The potential colonies with the features of *Aureobasidium* strains were selected. To detect the EPS-producing capacity, each strain was cultivated in the initial medium (12% sucrose, 0.3% yeast extract, 0.5% K_2_HPO_4_, 0.02% MgSO_4_·7H_2_O, 0.1% NaCl, and 0.06% (NH_4_)_2_SO_4_) at 28 °C, pH 6.5, and 180 rpm for 3 days [12,25]. The colonies were stored in a glycerol solution (20%) at −80 °C.

### 4.2. Strain Identification

Morphological characteristics of the target strain were shown on YPD and PDA plates at 28 °C for 2.5–8 days. The corresponding liquid media cultivated at 28 °C and 180 rpm for 16 h were used to observe the cell features using an optical microscope (Axio Lab. A1, Carl Zeiss, Oberkochen, Germany) with a 100 × oil immersion objective. Then, fresh cells harvested in the abovementioned liquid YPD medium were used to extract the genomic DNA using a Fungi Genomic DNA Extraction Kit (Solarbio, Beijing, China). To confirm the phylogenetic relationship between the target strain and other identified strains, the ITS sequence was amplified using primers ITS1 and ITS4, and sequenced [11]. A phylogenetic tree was accordingly constructed using MEGA 7.0 software (University of Kent, Canterbury, UK).

### 4.3. Optimization for High-Level EPS Production

To achieve high-level EPS production, the main influencing factors were optimized through the one-factor-at-a-time method at 28 °C and 180 rpm for 3 days [3,19]. These factors included carbon source (sucrose, maltose, glucose, fructose, or glycerol) and its optimal concentration (12–17%), (NH_4_)_2_SO_4_ (0.01–0.18%), K_2_HPO_4_ (0.1–0.9%), NaCl (0–0.5%), temperature (24–35 °C), and initial pH value (3.0–8.0).

### 4.4. Determination of EPS and DCW

The isolation of EPS and cells was performed using the previously reported method [47]. Briefly, the fermentation broth was heated (80 °C for 30 min) and centrifuged (12,000× *g* for 15 min) to remove cells and insoluble matter and obtain the supernatant and precipitate. Then, the precipitate was washed with distilled water and dried to obtain the DCW. The crude EPS from the supernatant was evaluated using pre-chilled anhydrous ethanol and dried to constant weight using an air-blast drying oven at 80 °C.

### 4.5. Purification and Identification of EPS

The EPS produced by strain ZH27 was purified using the Sevage method [10]. The crude EPS was used to prepare the EPS solution (5%). The proteins and gelatinous mass in the crude EPS were removed 5–7 times using the Sevage reagent (chloroform:n-butanol = 4:1). Purified EPS was obtained and then lyophilized, which was successively analyzed via pullulanase hydrolysis, TLC, and FTIR. Pullulanase is a typical debranching enzyme that is used to destroy α-1,6-glycosidic bonds of pullulan and produce maltotriose. The pullulanase (1000 ASPU/mL) (Yuanye Bio-Technology Co., Ltd., Shanghai, China) was added into the mixture containing the purified EPS or a standard pullulan solution (1.0%, *w*/*v*) and acetate buffer (50 mM and pH 4.5–5.0) with the proportions of 3.0:60:57 (*v*/*v*/*v*), which were subsequently treated at 60 °C for 15 min [13]. The contents of reducing sugars in hydrolysates were measured as stated below. Then, TLC was used to confirm the composition of hydrolysates using silica gel G-60, the developing solution (n-butanol/acetic acid/water, 8:2:2, *v*/*v*/*v*), and a coloring agent solution (concentrated sulfuric acid/anhydrous ethanol, 1:3, *v*/*v*) based on the described method with a slight modification [10]. Compared with one standard pullulan, the purified EPS was further characterized using FTIR by combining it with potassium bromide (100:1) and scanning over a range of 4000–400 cm^−1^.

### 4.6. Batch Fermentation for Pullulan Production

To achieve high levels of pullulan, 4 kinds of seed media, namely, SM (5.0% sucrose, 0.3% yeast extract, 0.5% K_2_HPO_4_, 0.02% MgSO_4_, 0.06% (NH_4_)_2_SO_4_, and 0.1% NaCl) [25], YPD, YPS (using sucrose to replace glucose), and potato dextrose broth (PDB), were optimized at 28 °C and 180 rpm. Moreover, both the incubation time (12–48 h) and innoculation size of the seed (5–10%) were also implemented at the same condition.

Batch fermentation for strain ZH27 was then conducted for pullulan production in a 7 L fermenter (Qizhi Biological Engineering Equipment Co., LTD, Guangzhou, China) under aeration rates of 4.5–5.5 L/min and rotation speeds of 180–220 rpm at 28 °C for 156 h. During fermentation, the culture was sampled at 12 h intervals to detect pullulan titers, DCW, reducing sugars, and total sugar contents. Meanwhile, the pullulan produced under the IC was adopted as the control. All the above tests were undertaken in triplicate.

### 4.7. Determination of Reducing Sugars, Total Sugars, Cell Morphology, Viscosity, and pH Values

The contents of reducing sugars and total sugars in fermentation broths were determined using a reducing sugar assay kit (BC0230) and total carbohydrate content assay kit (BC2710) purchased from Solarbio (Beijing, China), respectively. Moreover, changes in cell morphology under the OC and IC at different fermentation time points were monitored as described regarding the strain identification. The viscosity and pH value of the supernatant at the end were measured using a digital rotary viscometer (NDJ-5S, Shanghai, China) and a pH meter (PHSJ-5, Shanghai, China), respectively.

### 4.8. Detection of Transcriptional Expressions of the Genes Related to Pullulan Synthesis

The expressions of genes involved in the syntheses of pullulan and by-products and related regulatory factors were measured via qPCR using a TB Green Premix Ex Taq II (Tli RNaseH Plus) (TaKaRa, Dalian, China). First, the cells of strain ZH27 were harvested from batch fermentation (7 L) when cultivated for 60 h according to the time course of pullulan accumulation. The total RNAs of samples were extracted using an UNl Q-10 Column Trizol Total RNA Isolation Kit (Sangon Biotech, Shanghai, China) and were further used to synthesize cDNA using Revert Aid Premium Reverse Transcriptase (EP0733, Thermo Scientific, Waltham, MA, USA) and stored at −20 °C.

The qPCR primers of the genes involved in the above pathways are shown in Appendix A. The qPCR profile was performed under the following conditions: initial denaturation at 95 °C for 30 s, followed by 35 cycles of denaturation at 95 °C for 5 s, annealing at 60 °C for 30 s, and an elongation step at 72 °C for 20 s. The β-actin was selected as a reference gene for normalizing all genes, and the obtained results were quantified using the 2^−ΔΔCt^ method [31].

### 4.9. Statistical Analysis

All results in this study were derived from three distinct experiments, with each experiment executed in triplicate. Statistical analyses were conducted using one-way ANOVA through GraphPad Prism 9.0.0 (GraphPad Software Inc., San Diego, CA, USA). The statistical results are presented as mean ± standard deviation. Significance levels between the two groups were determined on the basis of probability values (*p*), where *p* < 0.05 (*) and *p* < 0.01 (**) were considered statistically significant.

## 5. Conclusions

A novel strain of *Aureobasidium melanogenum* ZH27 with a pullulan-producing ability was identified and characterized. Strain ZH27 produced 115.4 g/L pullulan with a yield of 0.77 g/g and a productivity of 0.87 g/L/h at 132 h using 15% sucrose under the OC via batch fermentation. The comparison of pullulan accumulation under the OC and IC indicates that swollen cells and large-volume vacuoles may be responsible for the high levels of pullulan production. During the accumulation of high pullulan titers by strain ZH27, a multitude of relevant genes (e.g., *UGT1*, *Ags2*, *GPP*, *TPS1*, fructoslytranserase genes, *GluK*, and *PGK*) involved in the syntheses of pullulan and by-products, as well as the glycolysis pathway, all exhibited significant upregulation. This finding suggests that by-products likely play a dual role in enhancing cell resistance to hyper-osmotic stress and facilitating pullulan synthesis. Therefore, strain ZH27 with a high pullulan-producing ability may emerge as a promising candidate for large-scale pullulan production.

## Figures and Tables

**Figure 1 ijms-25-00319-f001:**
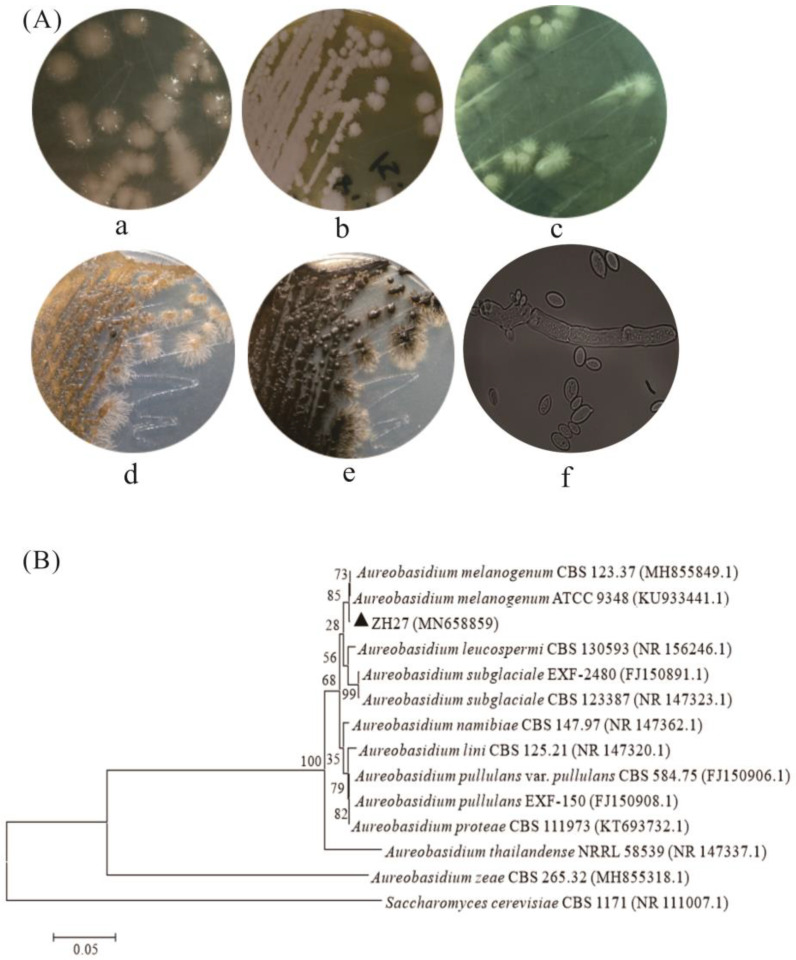
Morphological characteristics (**A**) and phylogenetic tree (**B**) of strain ZH27. (**A**(**a**,**b**)) represent the colonial morphologies of strain ZH27 growing on YPD plates for 2.5 and 5 days, respectively. (**A**(**c**–**e**)) are the colonial morphologies of strain ZH27 growing on PDA plates for 2.5, 5, and 8 days, respectively. (**A**(**f**)) shows the morphological features of strain ZH27 cells in liquid YPD after cultivating for 20 h (10 × 100 magnification). Diagram (**B**) illustrates the phylogenetic tree constructed using neighbor-joining analysis, and the triangle symbol means the target strain ZH27 can be easily found among all the strains on the tree. The bootstrap values are labeled based on 1000 replicates.

**Figure 2 ijms-25-00319-f002:**
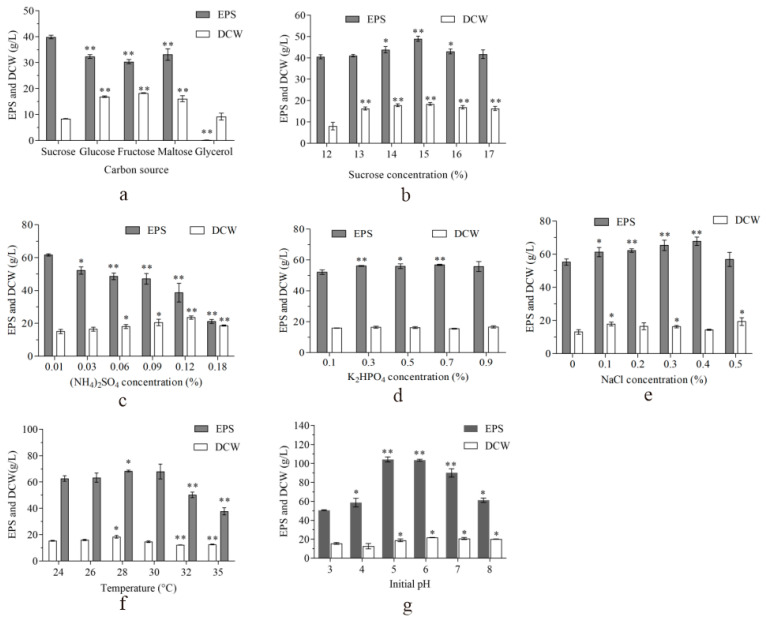
Optimization for EPS production. (**a**–**g**) represent the optimization of carbon source, sucrose, (NH_4_)_2_SO_4,_ K_2_HPO_4_, NaCl, temperature, and initial pH, respectively. “*” represents the probability value below 0.05, and “**” means the probability value below 0.01.

**Figure 3 ijms-25-00319-f003:**
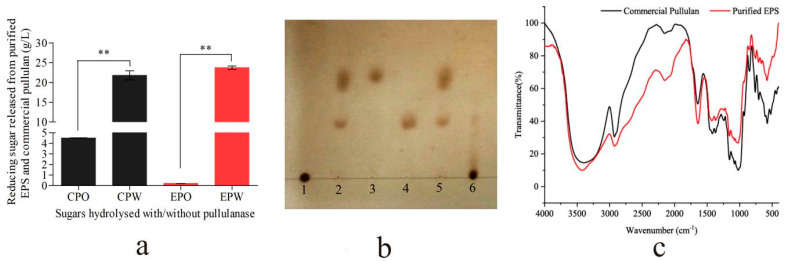
Structural identification of the purified EPS using hydrolysis, TLC, and Fourier-transform infrared spectroscopy (FTIR). (**a**) Reducing sugars released from the purified EPS and commercial pullulan. CPO and CPW marked with black denote the commercial pullulan (1%) hydrolyzed without and with pullanase, respectively. EPO and EPW marked with red represent the purified EPS (1%) treated without and with pullulanase, respectively. (**b**) Lines 1 and 2 show the purified EPS (1%) treated without and with pullulanase, respectively. Lines 3 and 4 represent glucose (1%) and maltotriose (1%), respectively. Lines 5 and 6 show commercial pullulan (1%) treated with and without pullulanase, respectively. (**c**) FTIR spectra of purified EPS and commercial pullulan. “**” represents the probability value below 0.01.

**Figure 4 ijms-25-00319-f004:**
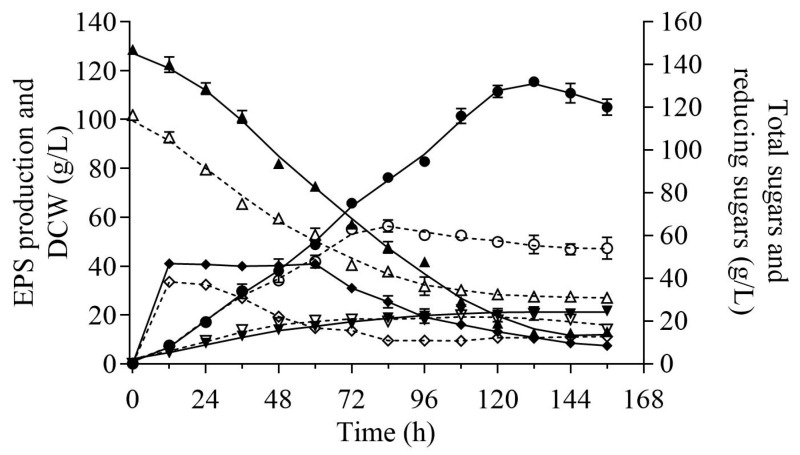
Time courses of pullulan production (○ and ●), DCW (▽ and ▼), and changes in reducing sugars (◊ and ♦) and total sugars (△ and ▲) during the 7 L batch fermentation of strain ZH27 under the initial condition (IC) (dash lines) and optimized condition (OC) (solid lines), respectively.

**Figure 5 ijms-25-00319-f005:**
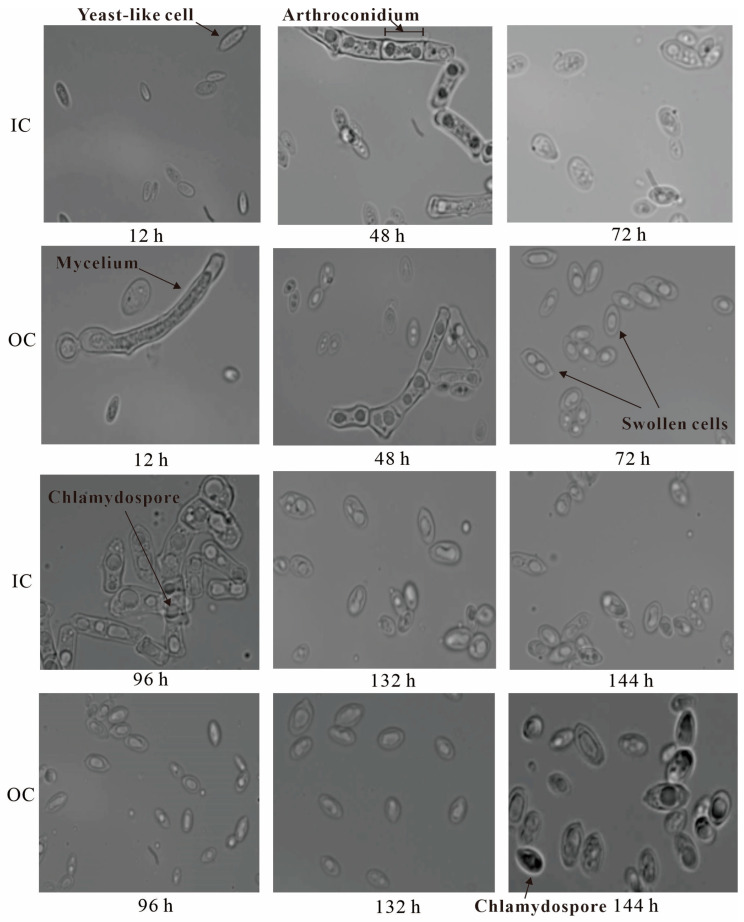
Cell morphological features of strain ZH27 cultivated under initial condition (IC) (12% sucrose) and optimized condition (OC) (15% sucrose) in the fermentation period of 12–144 h. The images were captured using the 100× oil immersion objective.

**Figure 6 ijms-25-00319-f006:**
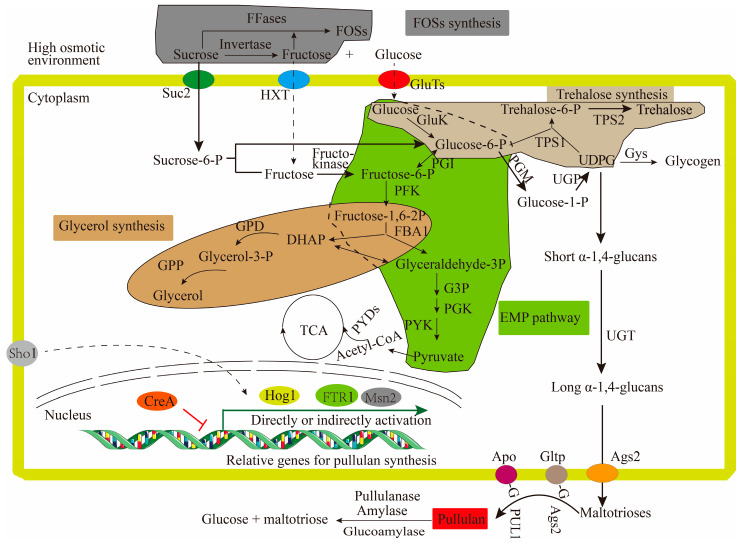
Sensing and utilization of sucrose by strain ZH27 under the high osmotic stress for producing pullulan. HXT, hexose transporter; Gys, glycogen synthase; DHPA, dihydroxyacetone phosphate. Other enzymes and regulatory factors are mentioned above in this study. Green and red arrows indicate activation and inhibition, respectively.

**Table 1 ijms-25-00319-t001:** Comparison of transcriptional levels of the genes involved in the syntheses of pullulan, trehalose, glycerol, and metabolism in strain ZH27 between the OC and IC.

Genes	Contents/Functions	Relative Gene Expressions of Strain ZH27 (%)		Changes in Gene Expressions under the OC
		IC	OC	
*GluT1*	Glucose transporter	100	70.9 ± 14.3 *	Downregulation
*GluT2*	Glucose transporter	100	1386.1 ± 485.8 **	Upregulation
*GluT3*	Glucose transporter	100	69.5 ± 3.3 **	Downregulation
*GluT4*	Glucose transporter	100	857.1 ± 126 **	Upregulation
*GluT5*	Glucose transporter	100	2894.9 ± 750.2 **	Upregulation
*SUC2*	Sucrose-proton symporter 2	100	1432.9 ± 364.1 **	Upregulation
*GluK*	Glucose kinase	100	1844.6 ± 499.4 **	Upregulation
*PGM*	Phosphoglucose mutase	100	118.6 ± 20.3	No significance
*UGP*	UDPG-pyrophosphorylase	100	215.4 ± 29.9 **	Upregulation
*UGT1*	UDP-glucosyltranferase	100	4742.8 ± 508.5 **	Upregulation
*PUL1*	Pullulan synthetase	100	438.9 ± 70.7 **	Upregulation
*Ags2*	Alpha-glucan synthase 2	100	367.5 ± 77.4 **	Upregulation
*Msn2*	Stress-responsive transcriptional activator	100	228.6 ± 13.5 **	Upregulation
*Apo*	Glycolipid transfer protein	100	153.5 ± 32.4 *	Upregulation
*Gltp*	Glycolipid transfer protein	100	377.7 ± 98.6 **	Upregulation
*CreA*	Regulation of gene transcription	100	222.9 ± 22 **	Upregulation
α-amylase gene	Hydrolysis of α-1,4-linkage	100	3938.8 ± 200 **	Upregulation
Glucoamylase gene	Hydrolysis of α-1,4- and α-1,6-linkages	100	2040.8 ± 281.3 **	Upregulation
Pullulanase gene	Hydrolysis of α-1,6-linkage	100	1262.1 ± 174 **	Upregulation
*TPS1*	α-trehalose-6-phosphate synthase	100	496.4 ± 50.1 **	Upregulation
*TPP*	Trehalose-6-phosphate phosphatase	100	278.7 ± 42.5 **	Upregulation
*GPP*	Glycerol-3-phosphatase	100	1341.7 ± 278.3 **	Upregulation
*GPD*	Glycerol-3-phosphate dehydrogenase	100	376.3 ± 56.4 **	Upregulation
*Hog1*	Hyper-osmotic glycerol	100	256.7 ± 33.7 **	Upregulation
*Sho1*	High osmolarity signaling protein	100	161.7 ± 4.0 **	Upregulation
*FF1*	Fructoslytranserase	100	373.4 ± 174.8 **	Upregulation
*FF2*	Fructoslytranserase	100	231.8 ± 56.7 **	Upregulation
*FF3*	Fructoslytranserase	100	215.2 ± 37.9 **	Upregulation
*FTR1*	Transcriptional factor of fructoslytranserase genes	100	333.1 ± 86.7 **	Upregulation
*PFK26*	Phosphofructo-2-kinase	100	257.8 ± 25.5 **	Upregulation
*FBA1*	Fructose-bisphosphate aldolase1	100	524.7 ± 60.5 **	Upregulation
*G3P*	Glyceraldehyde-3-phosphate dehydrogenase	100	523 ± 102.6 **	Upregulation
*PGK*	Phosphoglycerate kinase	100	484.1 ± 18.7 **	Upregulation
*PYK*	Pyruvate kinase	100	203.6 ± 28.9 **	Upregulation
*PYD*	Pyruvate decarboxylase	100	245.1 ± 64 **	Upregulation
*AreA*	Transcriptional activator of nitrogen	100	135.7 ± 20 *	Upregulation
*AreB*	Transcriptional repressor of nitrogen	100	85.3 ± 11 *	Downregulation

Notes: “*” represents the probability value below 0.05, and “**” means the probability value below 0.01.

**Table 2 ijms-25-00319-t002:** Pullulan derived from fermentation of *Aureobasidium* strains recently.

Strains	Fermentation		Production g/L (g/L/h)	Yield (g/g)	References
	Carbon	Mode and Conditions			
*A. pullulans* MG271838	Sucrose	Batch, 28 °C, 200 rpm, 168 h	37.6 (0.22)	0.63	[2]
*A. pullulans* H31	Sucrose	Batch, 28 °C, 200 rpm, 96 h	51.4 (0.54)	0.93	[3]
*A. melanogenum* 13-2	Sucrose	Batch, 28 °C, 250 rpm, 6.5 L/min, 120 h	78.1 (0.65)	0.56	[8]
*A. melanogenum* A4	Maltose	Fed-batch, 30 °C, 180 rpm, ~3 L/min, 120 h	122.3 (1.02)	0.40	[10]
*A. melanogenum* TN1-2	Sucrose	Batch, 28 °C, 250 rpm, 6.5 L/min, 132 h	114 (0.86)	0.81	[13]
*A. melanogenum* TN3-1	Glucose	Batch, 28 °C, 250 rpm, 6.5 L/min, 132 h	110.3 (0.84)	0.79	[18]
*A. Pullulans* M233-20 ^a^	Glucose	Batch, 28 °C, 500 rpm, 20 L/min, 144 h	162.3 (1.13)	0.82	[14]
*A. pullulans* LB83	SBH	Batch, 25.3 °C, 232 rpm, 96 h	25.2 (0.28)	0.48	[36]
*A. melanogenum* V6 ^a^/F44 ^a^	Sucrose	Batch, 28 °C, 180 rpm, 144 h	102.3 (0.7)/ 101.4 (0.7)	0.89/ 0.88	[38]
*A. melanogenum* AMY-PKS-11 ^a^	Glucose	Batch, 28 °C, 300 rpm, 6.5 L/min, 120 h	103.5 (0.86)	0.75	[39]
*A. melanogenum* DG41 ^a^	Glucose	Batch, 28 °C, 205 rpm, 8.0 L/min, 132 h	64.9 (0.49)	0.55	[25]
*A. Pullulans* BL06 ΔPMAs ^a^	Sucrose	Fed-batch, 28 °C, 500 rpm, 120 h	140.2 (1.17)	<0.5	[11]
*A. melanogenum* ZH27	Sucrose	Batch, 28 °C, 200–220 rpm, 4.5–5.5 L/min, 132 h	115.4 (0.87)	0.77	This study

Notes: SBH, sugarcane bagasse hydrolysate; ^a^, mutants.

## Data Availability

The data presented in this study are available on request from the corresponding authors.

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
