# Peer review of "Hyper-Production of Pullulan by a Novel Fungus of *Aureobasidium melanogenum* ZH27 through Batch Fermentation"

_ijms, 2023, doi:10.3390/ijms25010319_

Round 1

Reviewer 1 Report

Comments and Suggestions for Authors

The manuscript "Hyper-production of pullulan by a novel fungus of Aureobasidium melanogenum ZH27 through batch fermentation" presents the characteristics of a new isolate with increased production of pullulan. The subject matter is part of the scope of the journal, and research is constantly being carried out on the search for new strains with biotechnological potential.

Detailed notes:

Figure 1 A - figure f has a very dark background and is difficult to read - can the authors improve it?

The explanations of the drawings blend very closely with the text - I suggest that the authors use a smaller font, e.g. for lines 137-143.

Figure 4 - please brighten the background of the drawings.

Chapter 4.3 - please describe in more detail what breeding conditions were used.

Author Response

Dear Reviewer,

Thanks for your constructive suggestions.

We have extensively revised our manuscript, and have also improved our English expressions. All the changes are marked in red in the revised version. Below, I briefly outline the key points of our revision with respect to your comments.

Q1. Figure 1A-f has a very dark background and is difficult to read - can the authors improve it?

Response: Thanks for your suggestion. We have changed the background of Figure 1 A-f and improved its quality.

Q2. The explanations of the drawings blend very closely with the text - I suggest that the authors use a smaller font, e.g., for lines 137-143.

Response: Thanks for your suggestion. We have checked and changed the fonts of the explanations of Figures and Tables in the whole revised version.

Q3. Figure 4 - please brighten the background of the drawings.

Response: Good suggestion! We have brightened the background of Figure 4.

Q4. Chapter 4.3 - please describe in more detail what breeding conditions were used.

Response: Thanks for your suggestion. We have provided the more detailed information on the main factors and cultivating conditions for optimizing the high pullulan concentration.

Sincerely,

Dr. Qin-Qing Wang

Prof. Jiang-Hai Wang

Reviewer 2 Report

Comments and Suggestions for Authors

Review:

Hyper-production of pullulan by a novel fungus of Aureobasidium melanogenum ZH27 through batch fermentation

Qin-Qing Wang 1,2,†, *, Jia Lin 1,†, Qian-Zhi Zhou 1, Juan Peng 1, Qi Zhang 1, and Jiang-Hai Wang1,*

In the manuscript entitled "Hyper-production of pullulan by a novel fungus of Aureobasidium melanogenum ZH27 through batch fermentation", the authors describe how they isolated a new Aureobasidium melanogenum strain and investigated its pullulan production. For this purpose, media optimization was carried out using a one-factor-at-a-time method, the resulting medium of which is compared to the previous medium in a batch fermentation. Furthermore, the expression levels of different genes during pullulan production are compared with the non-optimized medium. In addition, the cell morphology using the two media will be investigated.

In sum, I believe the manuscript cannot be published in its present form. The abstract does not provide an overview of the subject matter and the introduction is, in my opinion, not detailed enough for this extensive topic. Different properties of pullulan, polyextremotolerant properties of Aureobasidium sp. strains, and their several by-products are not discussed. Furthermore, pullulan is already produced industrially and the current state of research is not presented. The results and discussion section is difficult to follow, there is frequent repetition, and conclusions that are not comprehensible are drawn. The material and methods section is also difficult to follow and lacks essential information, such as cultivation conditions. Furthermore, there are no detailed captions on figures and tables, references are often missing, and several conclusions that have been quoted could not be found in the corresponding literature. Furthermore, the language needs significant improvement. Some sentences cannot be understood as they are. Also, there are many typing errors.

Abstract:

The abstracts include only one sentence about the general topic. Moreover, it is not easy to understand. The abstract should be rewritten with the following structure:

1)      Background: Place the question addressed in a broad context and highlight the purpose of the study

2)      Methods: Briefly describe the main methods

3)      Results: Summarize the main findings of the article

4)      Conclusions: Give the main conclusions or interpretations

·        Line 21: Which process parameter increased by 105%?

·        Line 21: Unknown abbreviation: IM

·        Line 22-23: Using the optimized medium, cells containing 1-2 large vacuoles produce more pullulan. Using the not-optimized medium, cells contain 1 large vacuole. Please explain.

·        Line 24-25: This sentence is very difficult to understand: “Based on up-regulated gene expressions, the byproducts seemed likely to benefit cells for pullulan synthesis”. Please explain.

Introduction:

The introduction is only a tiny roundup of Aureobasidium sp. and pullulan. The authors do not introduce the topic in detail including the aim of this work.

·        Line 31: Please explain the “unique” structure of pullulan. How many repeating maltotriose subunits are present? How does that influence the molecular weight and the properties of these molecules? Is it important for the industry to have a specific pullulan molecule?

·        Line 35/36: Reference missing

·        Line 38-40: Reference missing

·        Line 41-43: Reference missing

·        Line 46-47: You write about extreme environments and robust strains. What does that mean? Is this advantageous for the industry?

·        Line 47: You are writing that it has been investigated that A. pullulans and A. melanogenum are the most robust pullulan producers and quote Wei et al. 2021. In my opinion, this review only summarized several pullulan producers. Please find another source that investigated the robustness of these strains or explain your quotation.

·        Line 50-51: How did they modify genes? Or do you mean genetic modification/ genetic engineering?

·        Line 49-53: You are writing about measures for enhanced pullulan production and quote 8-10. In the following sentence, you explain that these measures did not enhance pullulan production and quote sources 1, 9, and 11. Please explain.

·        Line 53: Why is it “urgent and indispensable” to find another pullulan producer? Pullulan is produced industrially and is well established in the industry. Please explain.

·        Line 53-55: Reference is missing

·        Line 56-57: What does “relatively high-level” mean? Please name process numbers. For example, titer, yield. and productivity of the best wildtype-based and mutant-based processes.

·        Line 56-57: Reference missing

·        Line 58-63: Explain the content of this work, but not the results.

Results

2.1. Screening and identification of EPS-producing strains

·        Line 67 and the following: You are writing about EPS titers before you explain your screening experiment and the identification of EPS

·        Line 71: Several strains were isolated and compared in the first step. Where are the results? You compared the strains based on pullulan titer but checked afterward only for ZH27 if it is EPS? Please explain.

·        Please introduce abbreviations like YPD, ITS, PDA, etc.

·        Line 74: Did you check if the colonies were surrounded or covered by EPS, or how do you identify that it is EPS?

·        Figure 1: Reorder the pictures from a-f.

·        Figure 1: Please insert a detailed caption

·        Figure 1-A-f: This is a low quality picture. It is not easy to see something

2.2. Optimization for EPS production

·        The medium optimization is an important part of this paper. Why are the graphs in the supplementary?

·        How did you perform your analysis of variance?

·        Line 95: missing reference

·        Line 99-102: You use sucrose concentrations between 12-17%. Compared to the other medium composition is this a very small interval. How did you choose your intervals?

·        Line 103: Name important numbers, for example, the highest titer, especially when the graph is in the supplementary

·        Line 105/106: Reference missing

·        Line 109: Why is 0.01% (NH4)2SO4 preferred? The decreasing EPS titers with increasing (NH4)2SO4 content while the initial sucrose content is constant, is only due to biomass formation. Consequently, more carbon is used for biomass and less for pullulan production. Please explain.

·        Line 110-112/ Figure S1-d: Including the standard deviation, there is no, very little difference in EPS titers. Please discuss.

·        Line 113-114: Reference missing

·        Line 119: You are writing about influencing cell growth by temperature and pH. What about high sucrose concentrations or different nitrogen concentrations?

·        Line 123: Several Aureobasidium sp. produce malate or polymalate. Is it possible that your medium lowers the pH during cultivation?

·        What is your final result? The medium composition. Please name it.

2.3. Structural analysis of EPS

·        You should introduce every chapter with some sentences

·        Is it necessary to perform a TLC in addition to the FTIR? Why?

·        Figure 2: detailed caption missing, maybe Lines 137-143?

·        Figure 2c: There is a big difference in transmittance between 2750 and 2000 cm-1. How do you explain that?

2.4 Comparison of pullulan production between IM and OM

·        You are performing several fermentations in a 7 L bioreactor but did not monitor or control anything. What about pH- and DO-control or offgas-analysis? Why do you use bioreactors without taking any advantage of it?

·        Line 159: A 7 Liter cultivation is, in my opinion, not a large-scale cultivation

·        Line 160-162: How did you investigate that 36 hours in YPS is the best cultivation time for your preculture? Did you check the biomass formation over time in your seed culture? 36 hours in a complex medium seems to be very long. Please explain.

·        Line 160-161: Why is an inoculum size of 6 % suitable? Including the standard deviation, your results in Figure S2-c are nearly the same.

·        Line 164: Explain total sugar and reducing sugar. Total sugar includes sucrose and FOS, while reducing sugars all monosaccharides are measured.

·        Line 168-169: The resulting supernatant viscosity using the IM medium is in the same range as honey, lubricants, or syrups. What does that mean for a process? What is the difference in the pullulan? Pullulan increases the viscosity of culture broth, but the pullulan content is doubled in the study, while the viscosity is only a fraction. Please explain.

·        Missing process parameters: growth rates, product-to-substrate yield, product-to-biomass yield, titer, specific productivity, space-time-yield, sugar consumption rates

2.5 Probable reasons for hyper-production of pullulan

·        Chapter introduction missing

2.5.1. Dynamic changes of cell morphology during batch fermentation

·        Figure 4: Microscope pictures are of low quality. It is not possible to make precise conclusions.

·        Figure 4: Detailed caption is missing

·        You should explain how you identify different cell types and mark them on the pictures

·        Line 185-188: There is no notable difference in 1-2 large vacuoles compared to 1 large vacuole. Moreover, in Line 197, you explain that the literature relates small vacuoles to higher pullulan titers. Please explain.

·        Line 192: But the pH using IM is even lower (2.97). So, why is this a reason for swollen cells chlamydospore formation using the OM and not while using the IM? Please explain.

2.5.2. Expression changes of the genes in pullulan synthesis and carbon metabolism

·        Line 201: Please explain FOS or describe them in your introduction.

·        Table 1: You are investigating a completely new strain. Which strain did you use for reference?

·        Table 1: Detailed caption is missing.

·        Line 231-232: Missing reference

Discussion

·        Line 246-248: Reference missing

·        Line 248-250: You are writing that Aureobasidium wild-type strains are usually low in pullulan production. Your reference (1, Wei et al. 2021) listed 5 strains with a titer over 100 g L-1 and a product-to-substrate yield of over 0.75. This is not “usually low”. Please explain.

·        Line 250-253: Your writing about the problem of genetic instability of mutants and quote (1) and (6) of your references. These authors did not write about this topic. Please explain. It is unclear why the industry needs wildtype strains for pullulan production.

3.1 ZH27 is a promising candidate for pullulan synthesis

·        Line 265: Please explain the term black yeast and the several by-products

·        Line 265: Reference missing

3.2. Robust pullulan synthesis by ZH27 after optimization

·        Why is it necessary to optimize the medium? There are several publications about pullulan production medium optimization.

·        Line 284-285: Reference missing.

·        Line 290-292: In my opinion, it is only due to biomass formation until 0.12% (NH4)2SO4. Please explain.

·        Line 297-298: Why is K2HPO4 the first factor influencing pullulan production? Please explain.

·        Line 301 and following: It is unclear what has been increased by how much. Titer, yield? Please describe this part better.

·        Line 304-307: Reference missing

·        Line 315-319: When the pH is so important for pullulan production, why didn’t you measure and control it in the bioreactor fermentation?

·        Line 324-325: High cost of what? Reference missing

·        Line 328: Your product-to-substrate yield is 0.77 g g-1. What is the theoretically possible yield? Aureobasidium strains can produce several by-products – how is it possible that ZH27 and other strains reach that high yields?

·        Line 333: You are writing that extracellular enzymes decompose your pullulan in your culture broth. Isn’t that a bad thing? Can the cells use pullulan as a carbon source? Even more so when it has been broken down into smaller units?

·        Line 337: Why is no melanin formation a good thing? Please explain.

3.3. Swollen cells, osmolytes, and high expression of related genes may be required for efficient pullulan production

·        Line 347: What is your control?

·        Line 351: Literature and you describe that pH strongly influences pullulan production. Why didn’t you observe that? Please explain.

·        Line 352-355: Do you have a reference for that? Sugumaran et al. (2017) state, “However, the exact morphology responsible for pullulan synthesis is still questionable.” However, your whole argumentation is based on bad microscope pictures and the difference between 1-2 large and 1 large vacuole. Please explain.

·        Line 359: Different citation styles

·        Line 361-363: References missing

·        Line 365-366: If high osmotic stress improves pullulan production, why is your pullulan production (Figure 1) constant over the complete cultivation period? According to the hypothesis, production should slow down with decreasing sugar content. Please explain.

·        Line 386: You are writing about reduced molecular weight and refer to table S3. In Table S3 is no molecular weight mentioned.

·        Line 397: Missing reference

·        Line 401-402: Which by-products? You used a completely new strain and did not measure any other products than pullulan.

·        Line 421-426: This should not be part of the discussion

·        Figure 5: Detailed caption is missing

Materials and methods

·        Abbreviations are not introduced

·        Cultivations are not described in detail: volumes, shake flask, MTPs, shaking diameter

·        Line 448: Detailed microscope settings are missing

·        Line 449: What do you mean with fresh cells? Cultivation conditions are missing.

·        Line 455: How do you know that these are the main influencing factors, reference? Why do you use a single-factor test and not a design of experiments?

·        Line 459-463: Please explain in detail. Why do you heat your fermentation broth? What is the weighing method?

·        Line 465: Explanation of crude EPS is missing

·        Line 471: How much pullulanase did you use?

·        Line 486: Incubation and cultivation details are missing

·        Line 487: Scaled up from what? A scale-up describes the process from a little bioreactor to a bigger one. Switching from a shaken flask or microtiter plate into the bioreactor is not only a question of the size. Please describe and discuss that.

·        Line 487: Describe your bioreactor in detail: pH-probe, DO-probe, which gas? Why do you use different aeration rates? Adjusting the dissolved oxygen content is more efficient by varying the stirring rate. Please explain.

·        Line 505: Why do you investigate your transcription level after 60 hours? The pullulan production is constant over time.

·        Line 517: Your analysis of variance was only performed for the triplicates in every experiment, right? Why didn’t you check if the results from different medium compositions differ significantly? Please explain.

Conclusion

·        Line 522: Please name the complete strain name

·        Maybe add the final product-to-substrate yield to your conclusion.

·        Line 528-529: How do you know that your strain is usable in large-scale fermentations? 7 Liter is not a large-scale fermentation. Please explain.

Comments on the Quality of English Language

The English use is poor.

Author Response

Responses to the Comments from Reviewer 2

Dear Reviewer,

Thanks for your constructive suggestions.

We have extensively revised our manuscript, and have also improved our English expressions. All the changes are marked in red in the revised version. Below, I briefly outline the key points of our revision with respect to your comments.

Abstract:

Q1. The abstracts include only one sentence about the general topic. Moreover, it is not easy to understand. The abstract should be rewritten with the following structure: Background, Methods, Results and Conclusions.

Response: Thanks for your good suggestion. We have rewritten the abstract in the revised version.

Q2. Line 21: Which process parameter increased by 105%?

Response: Thanks for your comments. The pullulan titer (115.4 g/L) obtained by A. melanogenum ZH27 at the optimized condition increased by 105% in comparison with that at the initial condition. The optimized conditions included the concentrations of sucrose, (NH4)2SO4, K2HPO4 and NaCl, temperature and initial pH. Obviously, it was the synergistic effect of the above-mentioned optimized process parameters for making pullulan titer increase by 105%.

Q3. Line 21: Unknown abbreviation: IM

Response: Thanks for your comment. We also noticed that using the initial medium or the control could not accurately express the context of production. Thus, we have redefined the initial condition as IC for strain ZH27 for pullulan producuction.

Q4. Lines 22-23: Using the optimized medium, cells containing 1-2 large vacuoles produce more pullulan. Using the not-optimized medium, cells contain 1 large vacuole. Please explain.

Response: Thanks for your suggestion. As we all know, Aureobasidium spp. are polymorphic fungi with five cell types, including yeast-like cells, arthroconidia, swollen cells, chlamydospores and mycelia. In addition, their cell types may change at different environments and stages in the period of pullulan production. In comparison with the initial condition (IC), higher sucrose concentration, higher C/N ratio and pH changes under the optimized condition (OC) could induce the changes of cell morphology and vacuoles of strain ZH27. The results in figure 4 show that most of cells in the OC changed from yeast cells into swollen cells with 1–2 large vacuoles; while the main cell types in the IC were swollen cells with a small amount of yeast-like cells, arthroconidia and mycelia. At the same time, swollen cells in the IC contained one large vacuole together with several small ones. Their difference was in the proportion of swollen cells and amount of large vacuoles. Therefore, we consider that swollen cells and the amount of large vacuoles in swollen cells might have a stronger and more stable ability to withstand the osmotic stress and require for pullulan synthesis in terms of the difference of cell morphology of strain ZH27 in the IC and OC. Moreover, these changes were likely related to the accumulation of trehalose and glycerol. In the future, we will focus on swollen cells and large vacuoles for exploring how they regulate pullulan synthesis.

Introduction:

Q5. The introduction is only a tiny roundup of Aureobasidium sp. and pullulan. The authors do not introduce the topic in detail including the aim of this work.

Response: Thanks for your suggestion. We have provided the more detailed information in the introduction in the revised version, in particular by-products, mainly including fructooligosaccharides (FOSs), trehalose and glycerol during pullulan synthesis as well as their influences to pullulan synthesis.

Q6. Line 31: Please explain the “unique” structure of pullulan. How many repeating maltotriose subunits are present? How does that influence the molecular weight and the properties of these molecules? Is it important for the industry to have a specific pullulan molecule?

Response: Thanks for your comments. As shown in Figure 1, the structure of pullulan consists of repeating maltotriose subunits that connected by α-(1®6) glycosidic bonds, and the subunit is composed of three glucose molecules that connected by two α-(1®4) glycosidic bonds. The ratio between α-(1®4) glycosidic and α-(1®6) glycosidic bonds is a definite value (2:1); and all the constituent molecule of pullulan is glucose. Thus, pullulan is a linear homopolysaccharide without branched chains, which is different from other polysaccharides such as starch and cellulose. The regular alternant of the two types of glycosidic bonds make pullulan possess favorable properties, including good water solubility, adhesive ability, structural flexibility and elasticity, film-forming ability, oxygen impermeability, biocompatibility and biodegradability. We have inclined to recognize the structure as “unique” one. On the other hand, we have used the “fundamental” structure to replace “unique” one.

Figure 1. Molecular structure of pullulan.

The polymerization degree of pullulan generally changes with different conditions. The molecular weight of pullulan is at the interval of 1.5×104 to 1.0×107 Dalton (Chen et al., 2019; Singh et al., 2019); and its average molecular weight is roughly 2.0×105 Dalton with approximate 480 maltotrioses (Shah et al., 2022). Generally, the more repeating maltotriose subunits, the greater the molecular weight and viscosity for pullulan. The molecular weight and size distribution of pullulan also play important roles in physicochemical performances, mechanical properties and degradation rates of pullulan-based materials. Importantly, the high molecular weight of pullulan makes it exhibit the high mechanical strength and low degradation rate, which may meet the requirements in drug delivery, gene targeting, tissue engineering and wound healing (Feng et al., 2022). Thus, we consider that a specific pullulan molecule is accurate and stable in the above applications.

Q7. Line 35/36: Reference missing

Response: Thanks for your suggestion. We have added the missing references [4,5] in the revised version.

Q8. Line 38-40: Reference missing

Response: Thanks for your suggestion. We have added the missing reference [4] in the revised version (Lines 45-46).

Q9. Line 41-43: Reference missing

Response: Thanks for your suggestion. This sentence was our summary on the basis of the studies on pullulan.

Q10. Line 46-47: You write about extreme environments and robust strains. What does that mean? Is this advantageous for the industry?

Response: Thanks for your comments. The pullulan producers, Aureobasidium strains could exist in some extreme environments. Here, we want to highlight that Aureobasidium strains can widely be found in nature and have a strong vitality. To our knowledge, unfavorable living environments such as deserts and oceans may prompt Aureobasidium strains to synthesize pullulan. If a target strain was obtained from an unfavorable extreme environment, the possibility will be improved for the industrial production of high-yield pullulan.

Q11. Line 47: You are writing that it has been investigated that A. pullulans and A. melanogenum are the most robust pullulan producers and quote Wei et al. 2021. In my opinion, this review only summarized several pullulan producers. Please find another source that investigated the robustness of these strains or explain your quotation.

Response: Thanks for your suggestion. We have added two other references (Chen et al., 2019; Chen et al., 2023) on Aureobasidium melanogenum A4 and Aureobasidium pullulans BL06. Both of them were screened and recognized as two relatively high pullulan-producing strains in comparison with the others screened from the same batch. At the same time, the two references have been added into the revised version.

Q12. Lines 50-51: How did they modify genes? Or do you mean genetic modification/ genetic engineering?

Response: Thanks for your comment. Indeed, we mean the genetic modification/genetic engineering. This process was carried out by homologous recombination (Chen et al., 2020). For instance, two homologous arms (5′-arm and 3′-arm) at the both ends of gene Ags2 were amplified with the corresponding restriction sites. The 5′-arm and 3′-arm of the gene were digested with the corresponding enzymes and ligated into the knockout plasmid pFL4A-NAT-loxp to form pFL4A-NAT-loxp-ΔAGS2. Then, the linear 3′-arm-loxp-PGK-NATpolyA- loxp-5′-arm fragments from the knockout vector were prepared by digesting them with the corresponding DNA restriction enzymes. Finally, the linear fragments were transformed into the competent cells of A. melanogenum P16 and the disruptants could be obtained on the double layer HCS plates at 28℃.

For over-expression, gene AGS2 without the introns was PCR-amplified with restriction sites and inserted into the expression plasmid pNTX13-NS-loxp to form pNTX13-NS-loxp-AGS2 using the same enzymes. The linear expression fragments could be obtained using the corresponding DNA restriction enzymes and transformed into the competent cells of different disruptants and wild-type strains P16 and A. melanogenum CBS105.22. Finally, the transformants over-expressing AGS2 could be obtained by the same knockout method.

Q13. Lines 49-53: You are writing about measures for enhanced pullulan production and quote 8-10. In the following sentence, you explain that these measures did not enhance pullulan production and quote sources 1, 9, and 11. Please explain.

Response: Thanks for your comment. This sentence means that several measures such as screening strains, condition optimization, mutagenesis, gene modification and substance addition could enhance pullulan production, but the degree of enhancement for pullulan production via these measures was no more than 35%. According to your suggestion, we have improved our expression in the revised version.

Q14. Line 53: Why is it “urgent and indispensable” to find another pullulan producer? Pullulan is produced industrially and is well established in the industry. Please explain.

Response: Thanks for your comment. Though some strains can be used for industrial production at present, their activities may gradually decrease, in particular the occurrence of functional degradation after long-term use, which will further affect the stability of industrialization. Therefore, mining new microbial resources for pullulan production is conducive to the stability and improvement of production performance.

Q15. Line 53-55: Reference is missing

Response: Thanks for your suggestion. We have added the missing reference in the revised version.

Q16. Line 56-57: What does “relatively high-level” mean? Please name process numbers. For example, titer, yield. and productivity of the best wildtype-based and mutant-based processes.

Response: Thanks for your suggestion. We have improved the related information in the revised version.

Q17. Line 56-57: Reference missing

Response: Thanks for your suggestion. We have added the missing reference in the revised version.

Q18. Line 58-63: Explain the content of this work, but not the results.

Response: Thanks for your comment. We have improved the related expressions in the revised version.

Results

Q19. Line 67 and the following: You are writing about EPS titers before you explain your screening experiment and the identification of EPS

Response: Thanks for your comment. Indeed, we adopted the general separation method of EPS (ethanol precipitation method) for obtaining the EPS titers of different strains. According to the different EPS titers, we screened a potential target strain for producing a relatively high EPS titer. Then, we identified the obtained EPS.

Q20. Line 71: Several strains were isolated and compared in the first step. Where are the results? You compared the strains based on pullulan titer but checked afterward only for strain ZH27 if it is EPS? Please explain.

Response: Thanks for your suugestion. We screened and obtained three EPS-producing strains, one produced 40 ± 1.2 g/L EPS, and the two others produced 30 ± 2.1 g/L EPS. Thus, we selected the strain (producing 40 ± 1.2 g/L EPS) as our target strain (i.e., strain ZH27) for performing further studies.

Q21. Please introduce abbreviations like YPD, ITS, PDA, etc.

Response: Thanks for your good suggestion. We have checked and improved the expressions in the revised version.

Q22. Line 74: Did you check if the colonies were surrounded or covered by EPS, or how do you identify that it is EPS?

Response: Thanks for your good suggestion. In fact, we have studied another strain A. melanogenum P16 (Ma et al., 2014; Wang et al., 2017; Kang et al., 2021), which produced pullulan in the liquid fermentation. At the same time, the colonies of strain P16 were surrounded or covered by pullulan (EPS). In this work, we selected potential strains with the producing-pullulan ability on the basis of their colony features.

Q23. Figure 1: Reorder the pictures from a-f.

Response: Thanks for your suggestion. We have reordered the pictures of Figure 1 in the revised version.

Q24. Figure 1: Please insert a detailed caption

Response: Thanks for your suggestion. We have inserted the detailed caption of Figure 1 in the revised version.

Q25. Figure 1-A-f: This is a low quality picture. It is not easy to see something

Response: Thanks for your suggestion. We have improved the quality of Figure 1 in the revised version.

2.2. Optimization for EPS production

Q26. The medium optimization is an important part of this paper. Why are the graphs in the supplementary?

Response: Thanks for your suggestion. We have provided some graphs on the medium optimization in the revised version.

Q27. How did you perform your analysis of variance?

Response: Thanks for your comment. We performed the data analysis using one-way ANOVA via GraphPad Prism 9.0.0 (GraphPad Software Inc., USA), and represented as mean ± standard deviation. Statistically significant differences between two groups based on the probability value (p) were considered as p < 0.05 (*) and p < 0.01 (**).

Q28. Line 95: missing reference

Response: Thanks for your suggestion. We have added the missing reference in the revised version.

Q29. Lines 99-102: You use sucrose concentrations between 12-17%. Compared to the other medium composition is this a very small interval. How did you choose your intervals?

Response: Thanks for your comment. Most of wild-type strains produce high pullulan titers at the sucrose concentrations of less than 15% (Jiang et al., 2018; Xue et al., 2019). Low sucrose concentrations obtain low pullulan titers (Hamidi et al., 2019). High sucrose concentrations would have the potential for achieving high pullulan titers, but pullulan titers will be reduced if sucrose concentrations are too high (Ma et al., 2014). On the other hand, high sucrose concentrations can increase the osmotic stress and affect cell growth and metabolite synthesis. Wild type strains with the pullulan-producing ability are sensitive to the changes of sucrose concentrations. Thus, we selected a small interval of sucrose concentrations for optimizing the conditions.

Q30. Line 103: Name important numbers, for example, the highest titer, especially when the graph is in the supplementary

Response: Thanks for your suggestion. We have checked and improved the highest pullulan titer in the revised version.

Q31. Line 105/106: Reference missing

Response: Thanks for your suggestion. We have added the missing reference in the revised version.

Q32. Line 109: Why is 0.01% (NH4)2SO4 preferred? The decreasing EPS titers with increasing (NH4)2SO4 content while the initial sucrose content is constant, is only due to biomass formation. Consequently, more carbon is used for biomass and less for pullulan production. Please explain.

Response: Thanks for your suggestion. Pullulan is induced to be produced under an adverse environment and can protect cells to withdraw the external environmental stress. High carbon concentrations and C/N ratios usually benefit for pullulan synthesis. Moreover, nitrogen catabolite repression may occur in pullulan production by Aureobasidium strains, which can inhibit pullulan synthesis in turn. In our previous study, genes AreA and AreB related to nitrogen catabolite repression in Aureobasidium strains were over-expressed (AreA after dephosphorylation) and knocked out (AreB) simultaneously, leading to the derepression of nitrogen catabolite repression (Kang et al., 2021). In this study, when the concentrations of (NH4)2SO4 were high, a majority of sucrose was used for biomass formation, and expressions of related enzymes were weak. On the other hand, nitrogen limitation made AreB inactivated and AreA activated, which would facilitate the carbon flux towards metabolite synthesis instead of biomass formation owing to activation of some enzymes and enhancement of ATP supply (Kang et al., 2021). Therefore, in the range of 0.01-0.18% (NH4)2SO4, 0.01% (NH4)2SO4 was recognized to be the preferred one.

Q33. Line 110-112/ Figure S1-d: Including the standard deviation, there is no, very little difference in EPS titers. Please discuss.

Response: Thanks for your suggestion. In comparison with the first EPS titer from 0.1% K2HPO4, the differences in EPS titers at 0.3-0.7% K2HPO4 were all significant using one-way ANOVA via GraphPad Prism 9.0.0.

Q34. Lines 113-114: Reference missing

Response: Thanks for your suggestion. We have added the missing reference in the revised version.

Q35. Line 119: You are writing about influencing cell growth by temperature and pH. What about high sucrose concentrations or different nitrogen concentrations?

Response: Thanks for your good suggestion. Indeed, temperature, pH, sucrose or nitrogen concentrations can all influence cell growth. High sucrose or nitrogen concentrations influenced cell growth (i.e., DCW) (see the optimization section of carbon source and (NH4)2SO4).

Q36. Line 123: Several Aureobasidium sp. produce malate or polymalate. Is it possible that your medium lowers the pH during cultivation?

Response: Thanks for your comment. Generally, one specific substance may be produced under a certain condition; and Aureobasidium strains produce polymalate at the occurrence of calcium carbonate. In the optimized medium, there was no calcium carbonate. Thus, malate or polymalate was not high. Generally, organic acids can be produced by microorganisms via glycolysis and tricarboxylic acid cycle metabolism. During the cultivation process, the pH value of the medium (fermentation broth) seemed to decline, due to the production of small organic acids including malate.

Q37. What is your final result? The medium composition. Please name it.

Response: Thanks for your suggestion. The final optimized medium at 28 ℃ and initial pH 5 was composed of 15% sucrose, 0.3% yeast extract, 0.3 % K2HPO4, 0.02% MgSO4·7H2O, 0.4% NaCl and 0.01% (NH4)2SO4.

2.3. Structural analysis of EPS

Q38. You should introduce every chapter with some sentences

Response: Thanks for your suggestion. We have checked and improved chapters in the revised version.

Q39. Is it necessary to perform a TLC in addition to the FTIR? Why?

Response: Thanks for your comment. In comparison with standard samples, the functional groups and glycosidic bonds in pullulan can be illustrated by FTIR, but it still needs to be identified because α-(1®4) glycosidic bonds can also be found in the other EPS. Thus, we did perform a study on the hydrolysates of EPS degraded by pullulanase on TLC plates for mutual verification.

Q40. Figure 2: detailed caption missing, maybe Lines 137-143?

Response: Thanks for your suggestion. We have added the detailed caption of Figure 2 in the revised version.

Q41. Figure 2c: There is a big difference in transmittance between 2750 and 2000 cm-1. How do you explain that?

Response: Thanks for your comment. The difference of FTIR spectra (2,750-2,000 cm-1) between the purified EPS and standard pullulan was obvious in our study. This may be attributed to the presence of some impurities in the purified EPS, which needs to be further clarified in the future. In addition, our obtained FTIR spectra were similar to the previous reports (Sheng et al., 2016; Yang et al., 2018; Liu et al., 2021; Chen et al., 2023), which were presented as below.

Figure 2. FTIR spectra of the purified EPS (red line) and a standard pullulan from Sigma (black line) (Liu et al., 2021)

Figure 3. FTIR spectra of pullulan produced by A. pullulans CGMCC1234 and a commercial pullulan (Sheng et al., 2016)

Figure 4. FTIR spectra of the purified EPS and a standard pullulan (Chen et al., 2023)

Figure 5. FTIR spectra of the purified EPS (red) by NCPS2016 and a standard pullulan (gray) from Sigma (Yang et al., 2018)

2.4 Comparison of pullulan production between IM and OM

Q42. You are performing several fermentations in a 7 L bioreactor but did not monitor or control anything. What about pH- and DO-control or offgas-analysis? Why do you use bioreactors without taking any advantage of it?

Response: Thanks for your valuable suggestion. We noted that the pH values decreased first and then rose at the later stage; while DO dropped during the process. We did not conduct an in-depth analysis. According to your suggestion, we will regulate pH and DO via our 7-L bioreactor for observing the effects on pullulan production and further analyzing the offgas.

Q43. Line 159: A 7 Liter cultivation is, in my opinion, not a large-scale cultivation

Response: Thanks for your suggestion. Here, we adopted the “larger-scale cultivations (7-liter)” in comparison with the shake-flask level. Thus, we have modified the related expressions in the revised version.

Q44. Lines 160-162: How did you investigate that 36 hours in YPS is the best cultivation time for your preculture? Did you check the biomass formation over time in your seed culture? 36 hours in a complex medium seems to be very long. Please explain.

Response: Thanks for your comment. As shown in Figure S1 in the revised version, the pullulan titers were determined for elevating the cultivation time of seed in YPS increasing, which were significantly higher at the cultivation time of seed in YPS at 24-48 h than that at 12 h. Obviously, when the seeds in YPS were cultivated for 36 h or 48 h, changes of their corresponding pullulan titers were found insignificantly and the pullulan titers from the seeds cultivating for 36 h or 48 h were recognized to reach the highest pullulan titer at the same time. Taking the time and energy consumption into account, the duration of 36 h in YPS was the best cultivation time for pullulan production. Yet we did not determine the biomass of seed culture. The aim of seed optimization was to obtain the best state of seed for pullulan production. This time is a little long for the fungus. The long time (36 h) for seed culture is probable due to a little initial inoculation using a ring of the seed solution from the seed tube.

Q45. Line 160-161: Why is an inoculum size of 6% suitable? Including the standard deviation, your results in Figure S2-c are nearly the same.

Response: Thanks for your comment. Indeed, the inoculum size is 5%. We have improved the description of Figure S1 (after revised) in the revised version.

Q46. Line 164: Explain total sugar and reducing sugar. Total sugar includes sucrose and FOS, while reducing sugars all monosaccharides are measured.

Response: Thanks for your comment. Total sugars include sucrose, FOSs, fructose, glucose, and pullulan; while reducing sugars include fructose, glucose, and FOSs.

Q47. Line 168-169: The resulting supernatant viscosity using the IM medium is in the same range as honey, lubricants, or syrups. What does that mean for a process? What is the difference in the pullulan? Pullulan increases the viscosity of culture broth, but the pullulan content is doubled in the study, while the viscosity is only a fraction. Please explain.

Response: Thanks for your comment. We found that the supernatant viscosity of strain ZH27 using the initial medium at 132 h was significantly higher than that in the optimized medium. As you said, it was like honey, lubricants, or syrups. During the fermentation process, the mobility of fermentation broth in the initial medium gradually became poor, and the fermentation broth was like thin pectin. In addition, the pullulan content was 48.8 ± 3.8 g/L in the initial medium at 132 h, which was added into Table S3 in the Supplementary Data. Whereas the supernatant viscosity of strain ZH27 under the optimized medium at 132 h was very low, due to the small molecular weight of pullulan. This type pullulan was produced under strong hydrolysis of α-amylase, glucoamylase and pullulanase, which might be over-expressed according to high gene expressions (Table 1).

Q48. Missing process parameters: growth rates, product-to-substrate yield, product-to-biomass yield, titer, specific productivity, space-time-yield, sugar consumption rates

Response: Thanks for your suggestion. We have checked and improved the process parameters in the revised version.

2.5 Probable reasons for hyper-production of pullulan

Q49. Chapter introduction missing

Response: Thanks for your suggestion. We have provided the missing chapter introduction in the revised version.

2.5.1. Dynamic changes of cell morphology during batch fermentation

Q50. Figure 4: Microscope pictures are of low quality. It is not possible to make precise conclusions.

Response: Thanks for your suggestion. We have improved the quality of Figure 5 (Figure 4 in the initial version) in the revised version.

Q51. Figure 4: Detailed caption is missing

Response: Thanks for your suggestion. We have added the caption of Figure 5 (Figure 4 in the initial version) in the revised version.

Q52. You should explain how you identify different cell types and mark them on the pictures

Response: Thanks for your suggestion. It has been investigated that Aureobasidium spp. strains have different cell types as described by Sugumaran and Ponnusami (2017), Liu et al. (2021) and Zeng et al. (2023). One cell in morphology is significantly different from others. According to experts’ experience and personal experimental basis, we identified the cell types of Aureobasidium spp. and marked them on the pictures of Figure 5 (Figure 4 in the initial version) in the revised version.

Q53. Line 185-188: There is no notable difference in 1-2 large vacuoles compared to 1 large vacuole. Moreover, in Line 197, you explain that the literature relates small vacuoles to higher pullulan titers. Please explain.

Response: Thanks for your comment. The quality of Figure 5 (Figure 4 in the initial version) in the revised version has been improved. “The most of swollen cells under the OC consisted of 1-2 large vacuoles and swollen cells with one vacuole were dominated after 48 h, whereas a great number of swollen cells under the IC contained one large vacuole together with several small ones (Figure 5).” This phenomenon appeared at the beginning of 48 h, and it could clearly be observed at 72 h and 132 h. In the literature of Xue et al.(2019), cell types were observed between the high pullulan-producing strain A. melanogenum TN3-1 (110 g/L pullulan) and low pullulan-producing strain A. melanogenum P16 (45 g/L pullulan) during fermentation. They found that the cells of strain TN3-1 contained larger cells, a thicker cell wall and a large amount of small vacuoles, while the cells of strain P16 had large vacuoles. They considered that small vacuoles in the cells of strain TN3-1 might be responsible for high resistance to high osmotic pressure and final high pullulan production. The difference between strain TN3-1 (from natural honey) and strain P16 or our novel strain of A. melanogenum ZH27 (from the mangrove ecosystem) may be attributed to differences of the long-term genetic evolution in different environments.

Q54. Line 192: But the pH using IM is even lower (2.97). So, why is this reason for swollen cells chlamydospore formation using the OM and not while using the IM? Please explain.

Response: Thanks for your comment. Firstly, we would like to highlight that “hyper-osmotic stress and low pH (the final pH 3.57) (Table S3) in OM compelled the strain ZH27 cells to form swollen cells earlier and more thoroughly than those in IM”, which did not mean that the lower pH of 2.97 in IM could not make the cells of strain ZH27 form swollen cells, but the formation time was later and only a part of yeast-like cells transferred into swollen cells. One study reported that yeast-like cells gradually differentiated to form swollen cells at low pH values (~4.5), where the growth and development of yeast-like cells were severely inhibited, yet swollen cells could adapt to this adverse environment (Zeng et al., 2022). It has been found that yeast-like cells can differentiate into swollen cells via the regulation of low pH values.

2.5.2. Expression changes of the genes in pullulan synthesis and carbon metabolism

Q55. Line 201: Please explain FOS or describe them in your introduction.

Response: Thanks for your suggestion. We have provided the detailed description on FOS in the introduction in the revised version.

Q56. Table 1: You are investigating a completely new strain. Which strain did you use for reference?

Response: Thanks for your comment. In this work, we found a novel strain, which was identified as A. melanogenum ZH27. An optimized medium for high pullulan production was obtained on the basis of the initial medium. In the transcriptional level, the expressions of genes related to the syntheses of pullulan and by-products of strain ZH27 under the optimized condition were compared with that under the initial condition which was used as the reference. It has not yet been compared with other strains. Your suggestion is very important for improving our work.

Q57. Table 1: Detailed caption is missing.

Response: Thanks for your suggestion. We have added the caption of Table 1 in the revised version.

Q58. Line 231-232: Missing reference

Response: Thanks for your suggestion. We have added the missing reference in the revised version.

Discussion

Q59. Line 246-248: Reference missing

Response: Thanks for your suggestion. We have added the missing reference in the revised version.

Q60. Line 248-250: You are writing that Aureobasidium wild-type strains are usually low in pullulan production. Your reference (1, Wei et al. 2021) listed 5 strains with a titer over 100 g L-1 and a product-to-substrate yield of over 0.75. This is not “usually low”. Please explain.

Response: Thanks for your comment. We mean that just the minority of wild-type Aureobasidium strains achieved relatively high pullulan (higher than 100 g/L) in the reference (Wei et al., 2021). Indeed, the majority of wild-type Aureobasidium strains produces low pullulan (≤ 78 g/L) (Ma et al., 2014; Jiang et al., 2019; Haghighatpanah et al., 2020). In comparison with the strains producing high pullulan, we consider that strain ZH27 may be considered as a candidate for pullulan production.

Q61. Line 250-253: Your writing about the problem of genetic instability of mutants and quote (1) and (6) of your references. These authors did not write about this topic. Please explain. It is unclear why the industry needs wildtype strains for pullulan production.

Response: Thanks for your comments. We have checked and improved the expression, including the genetic instability (Peng et al., 2020) in the revised version. Because engineered mutants can enhance the pullulan titer to some extent, but they encounter genetic instability causing strain degeneration with a greater probability. Thus, wild type strains with the ability for producing high pullulan titers are more suitable for industrial production of pullulan than engineered mutants.

3.1 strain ZH27 is a promising candidate for pullulan synthesis

Q62. Line 265: Please explain the term black yeast and the several by-products

Response: Thanks for your comment. In the sentence “As well-known black yeasts, Aureobasidium strains consist of five cell morphologies and synthesize several products”. Aureobasidium strains are dimorphic fungi, which have both major yeast cells and minor filamentous cells. Because most strains of the yeast-like fungi can synthesize melanin and accumulate it in their cell walls, Aureobasidium spp. are also called black yeasts (Wang et al., 2022). It is “several products” that include pullulan, polymalate, biosurfactant and liamocin (Meneses et al., 2017; Kang et al., 2022).

Q63. Line 265: Reference missing

Response: Thanks for your suggestion. We have added the missing reference in the revised version.

3.2. Robust pullulan synthesis by strain ZH27 after optimization

Q64. Why is it necessary to optimize the medium? There are several publications about pullulan production medium optimization.

Response: Thanks for your suggestion. Since strains isolated from different environments or even the same one generally possess different abilities. One strain could produce EPS under the optimized medium, but the others probably produced high EPSs titers under their own optimal media. Therefore, optimizing the medium should be performed for a new isolate. We have added the missing references in the revised version.

Q65. Line 284-285: Reference missing.

Response: Thanks for your suggestion. We have added the missing reference in the revised version.

Q66. Line 290-292: In my opinion, it is only due to biomass formation until 0.12% (NH4)2SO4. Please explain.

Response: Thanks for your suggestion. We have revised it in the revised version.

Q67. Lines 297-298: Why is K2HPO4 the first factor influencing pullulan production? Please explain.

Response: Thanks for your comment. Yang et al. (2018) found that K2HPO4·3H2O might be a key factor for EPS production. Moreover, Shen et al. (2019) indicated that K2HPO4 was the first important factor for influencing pullulan production by an orthogonal experiment. Thus, we consider that K2HPO4 was the first factor, and have checked and improved it in the revised version.

Q68. Line 301 and following: It is unclear what has been increased by how much. Titer, yield? Please describe this part better.

Response: Thanks for your comment. That indicated that the highest pullulan titer (67.8 ± 2.5 g/L) achieved at 0.4% NaCl increased by 11% in comparison with the control at 0.1% NaCl (Figure 2e). We have checked and revised this part in the revised version.

Q69. Line 304-307: Reference missing

Response: Thanks for your suggestion. We have added the missing reference in the revised version.

Q70. Line 315-319: When the pH is so important for pullulan production, why didn’t you measure and control it in the bioreactor fermentation?

Response: Thanks for your comment. We agree with you. In this work, we mainly focused on the fundamental research on strain ZH27, including screening and identification of strains, structural identification and optimization production of pullulan, and analyzing the probable reasons for high pullulan production. In the ongoing study, we will monitor and control the change of pH values in the bioreactor for industrial production.

Q71. Lines 324-325: High cost of what? Reference missing

Response: Thanks for your suggestion. We have checked and improved the expression (deleting the sentence) in the revised version.

Q72. Line 328: Your product-to-substrate yield is 0.77 g g-1. What is the theoretically possible yield? Aureobasidium strains can produce several by-products – how is it possible that strain ZH27 and other strains reach that high yields?

Response: Thanks for your constructive comment. A theoretically possible yield may be calculated by using response surface method or other design methods. Taking the cell growth, decrease of total sugars and uncertainty of the molecular weight of pullulan into consideration, it is difficult for us to calculate the theoretical yield. When Aureobasidium strains produced high pullulan titers using sucrose, they could also generate several by-products such as fructooligosaccharides (FOSs), glycerol and trehalose. FOSs formed outside the cells have investigated to reduce the environmental osmotic stress at an early fermentation stage and could also provide carbon source at the later stage for pullulan production (Sheng et al., 2016). Glycerol and trehalose formed inside the cell could resist the high osmotic stress (Chen et al., 2020). Based on the comparison of gene expressions by strain ZH27 between the optimized condition (OC) and initial condition (IC) , the three by-products were all up-expressed. In addition, the cell morphology and volume and number of vacuoles were also changed to be suitable cells producing pullulan. The changes of three by-products and pullulan produced by itself may make it possible to benefit cells for resisting the high osmotic stress and producing pullulan efficiently.

Q73. Line 333: You are writing that extracellular enzymes decompose your pullulan in your culture broth. Isn’t that a bad thing? Can the cells use pullulan as a carbon source? Even more so when it has been broken down into smaller units?

Response: Thanks for your comment. During the whole process of pullulan synthesis, many kinds of enzymes could be produced, including α-amylase, pullulanase and glucoamylase, which probably decomposed pullulan. Particularly, when carbon source was depleted, pullulan might be decomposed rapidly. For the purpose of pullulan production, it is not a good thing that extracellular enzymes decompose pullulan, resulting in the decrease of the pullulan titer in the culture broth. At the same time, the small molecules from decomposition could also be reused by cells, but this probably caused losses of material and energy in synthesis, degradation and re-synthesis processes. Therefore, the synthesized pullulan was decomposed, leading to reducing the pullulan titer, prolonging the fermentation time and increasing the production cost.

Q74. Line 337: Why is no melanin formation a good thing? Please explain.

Response: Thanks for your comment. Pullulan is a white product produced by the fermentation of Aureobasidium strains. During the fermentation, one by-product melanin is generally produced, leading to lowering the pullulan quality. Further, pullulan is difficult to separate from melan, which also increases the purification cost. In this study, strain ZH27 could efficiently produce pullulan without the formation of melanin in the fermentation period.

3.3. Swollen cells, osmolytes, and high expression of related genes may be required for efficient pullulan production

Q75. Line 347: What is your control?

Response: Thanks for your comment. Our control was the pullulan synthesis process of strain ZH27 under the IC. The related expressions have been improved in the revised version.

Q76. Line 351: Literature and you describe that pH strongly influences pullulan production. Why didn’t you observe that? Please explain.

Response: Thanks your comment. The sentence in the initial version was “This may be ascribed to the differences of nutrients and pH changes that influenced the differentiation of cells and vacuoles, leading to the difference of pullulan synthesis.” We considered that the differences in nutrients and pH changes influenced pullulan production. During the pullulan synthesis, the utilization rate of substrate sucrose by strain ZH27 under the OC was faster than that under the IC (Figure 4). The differential changes caused different levels of nutrients and pH values in both fermentation broths. As you suggested, we will focus on the effect of pH changes on pullulan production.

Q77. Line 352-355: Do you have a reference for that? Sugumaran et al. (2017) state, “However, the exact morphology responsible for pullulan synthesis is still questionable.” However, your whole argumentation is based on bad microscope pictures and the difference between 1-2 large and 1 large vacuole. Please explain.

Response: Thanks for your comment. We have improved the related expressions in the revised version. Before and after the sentence, we introduced the related studies on the main cell type (swollen cells) in pullulan synthesis (Zeng et al., 2023; and Zeng et al., 2022) and the effect of small vacuoles on pullulan production (Xue et al., 2019).

Q78. Line 359: Different citation style

Response: Thanks for your suggestion. We have improved the citation style in the revised version.

Q79. Line 361-363: References missing

Response: Thanks for your suggestion. We have added the missing references in the revised version.

Q80. Line 365-366: If high osmotic stress improves pullulan production, why is your pullulan production (Figure 1) constant over the complete cultivation period? According to the hypothesis, production should slow down with decreasing sugar content. Please explain.

Response: Thanks for your comment. The result in Figure 1 was not involved in constant pullulan production over the complete cultivation period. The sentence in Lines 365-366 (in the initial version) mean that high pullulan titers are gradually produced under the high concentrations of carbon source, which simultaneously causes a high osmotic stress. In this case, the genes related to syntheses of pullulan and by-products in Aureobasidium cells would make changes. Thus, we have improved the expression in the revised version.

Q81. Line 386: You are writing about reduced molecular weight and refer to table S3. In Table S3 is no molecular weight mentioned.

Response: Thanks for your comments. The viscosity change of pullulan is positively correlated with its molecular weight. In the initial version, we used the viscosity to reflect its molecular weight. Thus, we have improved the expression in the revised version.

Q82. Line 397: Missing reference

Response: Thanks for your suggestion. We have checked and added the missing reference in the revised version.

Q83. Line 401-402: Which by-products? You used a completely new strain and did not measure any other products than pullulan.

Response: Thanks for your comment. The by-products denote a small number of products produced by Aureobasidium. In the high osmotic stress condition of producing high pullulan titer, the by-products mainly include fructooligosaccharides (FOSs), glycerol and trehalose according to the previous studies (Hohmann 2015; Jiang et al., 2018; Xue et al., 2019; Chen et al., 2020). In this study, we did not directly determine the above-mentioned by-products, but the expressions of genes involving in their syntheses were all detected. The results indicate the up-regulation of related genes, suggesting that it is probable to produce higher titers under the optimized condition than those under the initial condition.

Q84. Line 421-426: This should not be part of the discussion

Response: Thanks for your suggestion. We have accepted your advice and deleted this part in the revised version.

Q85. Figure 5: Detailed caption is missing

Response: Thanks for your suggestion. We have provided the detailed caption in the revised version.

Materials and methods

Q86. Abbreviations are not introduced

Response: Thanks for your suggestion. All abbreviations such as YPD and PDA were provided in the Results.

Q87. Cultivations are not described in detail: volumes, shake flask, MTPs, shaking diameter

Response: Thanks for your suggestion. We have checked and added the detail in the revised version.

Q88. Line 448: Detailed microscope settings are missing

Response: Thanks for your suggestion. We have checked and added the detail in the revised version.

Q89. Line 449: What do you mean with fresh cells? Cultivation conditions are missing.

Response: Thanks for your comment. The fresh cells were the strain cultivated in the logarithmic growth phase, where the cells grew very fast. The missing cultivation conditions were also added in the revised version.

Q90. Line 455: How do you know that these are the main influencing factors, reference? Why do you use a single-factor test and not a design of experiments?

Response: Thanks for your comment. The composition and contents of the medium should be identified and we used the single-factor test for evaluating the pullulan titer significantly. The design of experiments is a good suggestion. We will learn to use the design of experiments for further improving the pullulan titer.

Q91. Lines 459-463: Please explain in detail. Why do you heat your fermentation broth? What is the weighing method?

Response: Thanks for your comment. We have cited one reference on the isolation method of EPS and cells (Sheng, et al., 2016). The fermentation broth was heated at 80 ℃ for 30 min and then centrifuged (12,000 × g, 15 min) for removing cells and insoluble matters. The EPS was evaluated from the supernatant using pre-chilled anhydrous ethanol and dried to constant weight using an air-blast drying oven at 80℃.

Q92. Line 465: Explanation of crude EPS is missing

Response: Thanks for your comment. We have provided the missing explanation on crude EPS, i.e., the crude EPS was obtained from the supernatant using pre-chilled anhydrous ethanol and dried to constant weight using an air-blast drying oven at 80℃.

Q93. Line 471: How much pullulanase did you use?

Response: Thanks for your comment. We added pullulanase (1000 ASPU/mL) into the mixture containing the purified EPS or a standard pullulan solution (1.0%, w/v) and acetate buffer (50 mM, pH 4.5–5.0) with the proportion of 3.0 : 60 : 57 (v/v/v). In this work, 0.03 mL pullulanase was used. Thus, the final added pullulanase concentration was 25 ASPU/mL.

Q94. Line 486: Incubation and cultivation details are missing

Response: Thanks for your comment. We have checked and added the detail in the revised version.

Q95. Line 487: Scaled up from what? A scale-up describes the process from a little bioreactor to a bigger one. Switching from a shaken flask or microtiter plate into the bioreactor is not only a question of the size. Please describe and discuss that.

Response: Thanks for your suggestion. We have checked and added the detail in the revised version.

Q96. Line 487: Describe your bioreactor in detail: pH-probe, DO-probe, which gas? Why do you use different aeration rates? Adjusting the dissolved oxygen content is more efficient by varying the stirring rate. Please explain.

Response: Thanks for your comment. In the bioreactor, we have done the experiment using the natural initial pH as the optimization at the shake flask level. Some changes during the fermentation process exhibited at the aeration rates of 4.5-5.5 L/min. Your suggestion on adjusting dissolved oxygen is very important for us to enhance pullulan production. We will further emphasize the downstream engineering for pullulan production in a larger scale.

Q97. Line 505: Why do you investigate your transcription level after 60 hours? The pullulan production is constant over time.

Response: Thanks for your comment. It can be seen in Figure 4 in the revised version that the pullulan titer by strain ZH27 under the optimized condition increased fast at the fermentation time of 60 h; and it was also in the rising stage of pullulan accumulation by strain ZH27 under the initial condition even though it was not very fast. In fact, the time of producing differential pullulan titers by strain ZH27 and cells growing increasingly could be selected. The time of 60 h did accords with the requirement.

Q98. Line 517: Your analysis of variance was only performed for the triplicates in every experiment, right? Why didn’t you check if the results from different medium compositions differ significantly? Please explain.

Response: Thanks for your comments. In this study, we performed the comparison of pullulan production, cell growth, total sugars, reducing sugars and transcriptional levels of the genes between the optimized and initial conditions. On the other hand, another medium may be more suitable for strain ZH27 to produce pullulan, but it needs to be further optimized. We consider that different media may cause different pullulan production. For the further work, we can also try other media for improving pullulan production according to your suggestion.

Conclusion

Q99. Line 522: Please name the complete strain name

Response: Thanks for your suggestion. We have provided the complete strain name in the revised version.

Q100. Maybe add the final product-to-substrate yield to your conclusion.

Response: Thanks for your suggestion. We have added the yield in the revised version.

Q101. Lines 528-529: How do you know that your strain is usable in large-scale fermentations?

Response: Thanks for your comment. We have improved the related expressions, i.e., strain ZH27 may be a target strain for exploring pullulan titer, production costs and large-scale production.

Q102. Liter is not a large-scale fermentation. Please explain.

Response: Thanks for your suggestion. We have changed the related expression on large-scale fermentation.

Sincerely,

Dr. Qin-Qing Wang

Prof. Jiang-Hai Wang

Reviewer 3 Report

Comments and Suggestions for Authors

This is a fine and extensive work on pullulan production by a new strain of Aurobasidium melanogenum ZH27, reaching really high titers over 115 g/L out of 130 g/L total sugars. It is reasonably well-written but for some expressions and words that need to be revised (please, see attached pdf files). In my opinion, is a fine work for IJMS, though it needs some improvements:

1)In this work there is not a proper statistical optimization. It will be preferible to use "best medium" or "enhanced medium" (BM or EM) instead of "optimized medium" (OM). Please, perform the adequate changes.

2) Please, indicate the byproducts for each batch. Moreover, what is the difference in byproducts between OM and IM? More discussion on byproduct effect on pullulan production is needed, once they are defined.

3) The assertion at the end of the discussion section does not seem to be justified by the results here presented. No literature is mentioned to support it. Consider its removal, in consequence.

4) Conclusion: I think it is exactly the contrary: the cell morphology is due to high pullulan production.

See typos in both the main manuscript (attached) and the supplementary material, please: there are some of them. In the supplementary material, the authors should provide tables with precise composition of broths OM and IM.

Comments on the Quality of English Language

Author Response

Responses to the Comments from Reviewer 3

Dear Reviewer,

Thanks for your constructive suggestions.

We have extensively revised our manuscript, and have also improved our English expressions. All the changes are marked in red in the revised version. Below, I briefly outline the key points of our revision with respect to your comments.

  1. In this work there is not a proper statistical optimization. It will be preferible to use "best medium" or "enhanced medium" (BM or EM) instead of "optimized medium" (OM). Please, perform the adequate changes.

Response: Thanks for your suggestion. We mainly used the one-factor-at-a-time method for optimizing the fermentation conditions, including culture media. According to your suggestion, the suitable optimized medium was named as enhanced medium (EM) and the optimized fermentation condition was named as OC. In addition, the media for producing pullulan before and after optimization were shown at the end of 2.2 Optimization for EPS production and 4.1 Sample collection and screening for EPS-producing strains in the revised version.

  1. Please, indicate the byproducts for each batch. Moreover, what is the difference in byproducts between OM and IM? More discussion on byproduct effect on pullulan production is needed, once they are defined.

Response: Thanks your good suggestion. The comparison of byproducts produced by strain ZH27 in EM and IM during pullulan synthesis and the effects of byproducts on pullulan synthesis should be considered. In the current study, we determined the changes of gene expressions related to the syntheses of byproducts between EM and IM, and found that most of related genes up-expressed significantly in EM, probably implying the increase of the synthesized byproducts. In our further study, we will mainly focus on the effects of byproducts on pullulan synthesis.

  1. The assertion at the end of the discussion section does not seem to be justified by the results here presented. No literature is mentioned to support it. Consider its removal, in consequence.

Response: Thanks for your good suggestion. We have checked and improved the expression in the revised version.

  1. Conclusion: I think it is exactly the contrary: the cell morphology is due to high pullulan production.

Response: Thanks for your comment. According to the current knowledge on pullulan production by different cell types of Aureobasidium spp., it is probable that cell morphology may be related to high pullulan production, due to the self-protection of pullulan; while there are some other important factors that probably influence cell morphology.

We consider that the high osmotic stress, nutrients and pH values may influence the cell differentiation, leading to the difference of pullulan synthesis in a high pullulan-producing environment. Some gene expressions could be compelled to change under a high osmotic stress (Wei et al., 2021). In addition, the cell types such as conidia can also be affected by changes of osmotic pressures in the environment (Zhao et al., 2012; Liao et al., 2022). It was also found that the osmotic stress prevented the differentiation of yeast-like cells to mycelia and chlamydospores (Liu et al., 2019). Liu et al.(2019) also found that A. pullulans NG could grow well as yeast-like cells under nutrient-rich conditions but changed to mycelia and chlamydospores when nutrients became poor. Moreover, pH could also affect cell morphology and pullulan yield. As core cells, yeast-like cells mainly existed at the beginning of cultivation (pH ≥ 6.0) and differentiated into swollen cells via the regulation of lower pH (Zeng et al., 2022).

  1. See typos in both the main manuscript (attached) and the supplementary material, please: there are some of them. In the supplementary material, the authors should provide tables with precise composition of broths OM and IM.

Response: Thanks for your suggestion. The enhanced medium composition was provided at the end of section 2.2. Optimization for EPS production, and the initial medium composition was provided at the end of section 4.1. Sample collection and screening for EPS-producing strains.

Sincerely,

Dr. Qin-Qing Wang

Prof. Jiang-Hai Wang

Round 2

Reviewer 2 Report

Comments and Suggestions for Authors

Thank you for revising your manuscript.

The discussion on pH seems a bit out of the blue, as it appears that no online pH measurement was used for control. Please explain in detail.

The first version was of low quality. Why did the reviewers first had to put a lot of effort in before the quality was increased? This is not nice to the the peers who have to do it. Still, the text is having issues like long sentences and grammar. You may want to use a editing firm or at least a software to further improve your manuscript.

Example: Moreover, a number of critical genes involved in syntheses of pullulan and by-products and glycolysis were almost all up-regulated under the OC, indicating that by-products probably help cells to resist hyper-osmotic stress and facilitate pullulan synthesis. Too long, to unspecific, nothing the reader can learn from. What is a critical gene? What is the plural of pullulan synthesis mean? Almost?! Pullulan (by-product) helps by-product synthesis? May be an experiment with far less carbon source but under the same osmotic conditions could elucidate the mechanism proposed.

Pullulan is an industrial product, hence all the sentences of low performance etc please rewrite or just delete. Hence, these measures still can not meet the industrial production of pullulan in terms of its efficiency and quality. This sentence as an example. Which measures? meet industrial production? As written in the text, pullulan is produced by firms, they will not have genetic instabilities etc. You cite studies with better performance than your strain. High increase is not a feature, if the final result is not better. Please be modest. You do not have to judge the other studies, just present what was reported. Figure 1 is not intuitive, as it is a mixture of different experiments and magnifications on solid and liquid media. It looks good, but as soon as one tries to understand the reason an the information content it gets difficult. Why are there subheaders with a super small font? Why A-a? May be A.1?

Comments on the Quality of English Language

The textual presentation is improved, but still requires more work (see also above).